# Bayesian Optimization via Continual Variational Last Layer Training

**Paul Brunzema**[1]**, Mikkel Jordahn**[2]**, John Willes**[3]**, Sebastian Trimpe**[1]**,
Jasper Snoek**[4]**, James Harrison**[4]
[1]RWTH Aachen University, [2]Technical University of Denmark, [3]Vector Institute,
[4]Google DeepMind
Contact: `brunzema@dsme.rwth-aachen.de`, `jamesharrison@google.com`

## Abstract

Gaussian Processes (GPs) are widely seen as the state-of-the-art surrogate models for Bayesian optimization (BO) due to their ability to model uncertainty and their performance on tasks where correlations are easily captured (such as those defined by Euclidean metrics) and their ability to be efficiently updated online. However, the performance of GPs depends on the choice of kernel, and kernel selection for complex correlation structures is often difficult or must be made bespoke. While Bayesian neural networks (BNNs) are a promising direction for higher capacity surrogate models, they have so far seen limited use due to poor performance on some problem types. In this paper, we propose an approach which shows competitive performance on many problem types, including some that BNNs typically struggle with. We build on variational Bayesian last layers (VBLLs), and connect training of these models to exact conditioning in GPs. We exploit this connection to develop an efficient online training algorithm that interleaves conditioning and optimization. Our findings suggest that VBLL networks significantly outperform GPs and other BNN architectures on tasks with complex input correlations, and match the performance of well-tuned GPs on established benchmark tasks.

## 1 Introduction

Bayesian optimization (BO) has become an immensely popular method for optimizing black-box functions that are expensive to evaluate and has seen large success in a variety of applications (Garnett et al., 2010; Snoek et al., 2012; Calandra et al., 2016; Marco et al., 2016; Frazier & Wang, 2016; Berkenkamp et al., 2017; Chen et al., 2018; Neumann-Brosig et al., 2019; Griffiths & Hernández-Lobato, 2020; Colliandre & Muller, 2023). In BO, the goal is to optimize some black-box objective $f \colon \mathcal{X} \to \mathbb{R}^K$ (where $\mathcal{X} \subseteq \mathbb{R}^D$) in as few samples as possible whilst only having access to sequentially sampled, potentially noisy, data points from possibly multiple objectives.

Gaussian processes (GPs) have long been the de facto surrogate models in BO due to their well-calibrated uncertainty quantification and strong performance in small-data regimes. However, their application becomes challenging in high-dimensional, non-stationary, and structured data environments such as drug-discovery (Colliandre & Muller, 2023; Griffiths & Hernández-Lobato, 2020) and materials science (Frazier & Wang, 2016). Here, often prohibitively expensive or bespoke kernels are necessary to capture meaningful correlations between data points. Furthermore, the scaling of GPs to large datasets typically associated with high-dimensional spaces can be limiting—especially if combined with online hyperparameter estimation. To address these challenges, integrating Bayesian Neural Networks (BNNs) into BO as alternative surrogate models has gained increasing attention (Snoek et al., 2015; Springenberg et al., 2016; Foldager et al., 2023; Li et al., 2024). While BNNs inherently scale with data, challenges like efficient conditioning on new data and consistency across tasks persist. Moreover, the performance of BNNs on many tasks has so far not matched the performance of GPs due to issues like under-fitting (Li et al., 2024; Wenzel et al., 2020; Ovadia et al., 2019; Izmailov et al., 2021).

In this work, we develop an approach for surrogate modeling in Bayesian optimization that combines the strengths of GPs (efficient conditioning, strong predictive performance, simple and effective

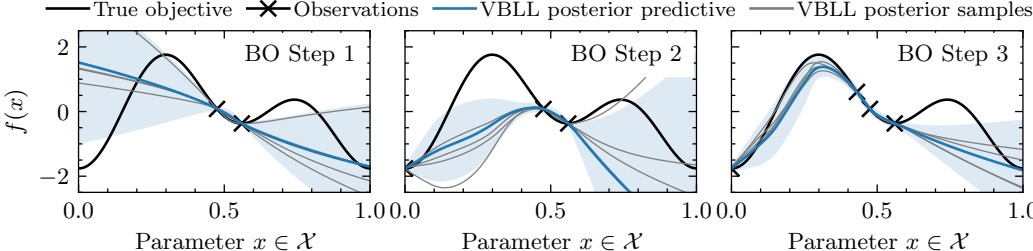

Figure 1: Variational Bayesian last layer model as a surrogate model for BO on a toy example. The VBLL model can capture *in-between* uncertainty and analytic posterior samples are easily obtained through its parametric form making it a suitable surrogate for BO.

uncertainty quantification) with the strengths of BNNs (scalability and ability to handle highly non-Euclidean correlation structures). Our approach builds on Variational Bayesian Last Layer (VBLL) neural networks (Harrison et al., 2024), which are similar to *parameteric* GPs with a learned kernel.

**Contributions:** Our main contributions and findings are:

- We show our VBLL surrogate model can outperform a wide variety of baselines on a diverse set of problems, including those with discrete inputs and multi-objective problems.
- We present a connection between model conditioning (in a Bayesian sense) and model optimization, and develop an efficient online optimization scheme that exploits this connection.
- We experimentally connect the hyperparameters of the VBLL model to problem features, such as stochasticity, and discuss implications for other BNN models.

## 2    RELATED WORK AND BACKGROUND

Various forms of Bayesian or partially-Bayesian neural networks have been explored for BO, including mean field BNNs (Foldager et al., 2023), networks trained via stochastic gradient Hamiltonian Monte Carlo (Springenberg et al., 2016; Kim et al., 2021), and last layer Laplace approximation BNNs (Kristiadi et al., 2021; Daxberger et al., 2021). Li et al. (2024) find that infinite-width BNNs (I-BNNs) (Lee et al., 2017; Adlam et al., 2021; 2023) perform particularly well especially on high-dimensional, non-stationary and non-Euclidean problems, a setting for which standard GP priors are inappropriate or not well specified, and for which designing suitable kernels is difficult.

While BNNs are promising, they have often proven to be challenging to train and complex to use in practice. Bayesian last layer networks—which consider uncertainty only over the output layer—provide a simple (and often much easier to train) partially-Bayesian neural network model (Snoek et al., 2015; Azizzadenesheli et al., 2018; Harrison et al., 2018; Weber et al., 2018; Riquelme et al., 2018; Fiedler & Lucia, 2023). Concretely, the standard model for regression with Bayesian last layer networks and a one-dimensional[1] output is

$$y = \boldsymbol{w}^\top \boldsymbol{\phi_\theta}(\boldsymbol{x}) + \varepsilon, \tag{1}$$

where $\boldsymbol{w} \in \mathbb{R}^m$ is the last layer for which Bayesian inference is performed, and $\boldsymbol{\phi_\theta}$ are features learned by a neural network backbone with (point estimated) parameters $\boldsymbol{\theta}$. The noise $\varepsilon \sim \mathcal{N}(0, \sigma^2)$ is assumed to be independent and identically distributed.

With this observation model, fixed features $\boldsymbol{\phi_\theta}$, and a Gaussian prior on the weights as $p(\boldsymbol{w}) = \mathcal{N}(\bar{\boldsymbol{w}}_0, S_0)$, posterior inference for the weights is analytically tractable via Bayesian linear regression. In particular, Bayesian linear regression is possible via a set of recursive updates in the natural

---

[1]For multi-variate modeling, we assume each element of $\varepsilon$ is independent, and so the $N$-dimensional problem is separable into $N$ independent inference problems with shared features $\boldsymbol{\phi_\theta}$. Relaxing this assumption is possible and leads to a matrix normal-distributed posterior.

parameters

$$\boldsymbol{q}_t = \sigma^{-2}\boldsymbol{\phi}_t y_t + \boldsymbol{q}_{t-1} \tag{2}$$

$$S_t^{-1} = \sigma^{-2}\boldsymbol{\phi}_t\boldsymbol{\phi}_t^\top + S_{t-1}^{-1} \tag{3}$$

where $\bar{\boldsymbol{w}}_t = S_t\boldsymbol{q}_t$ is the vector of precision-mean and $\boldsymbol{\phi}_t := \boldsymbol{\phi}_{\boldsymbol{\theta}}(\boldsymbol{x}_t)$. For dataset $\mathcal{D}_T := \{(\boldsymbol{x}_t, y_t)\}_{t=1}^T$, this recursive update yields posterior $p(\boldsymbol{w} \mid \mathcal{D}_T) = \mathcal{N}(\bar{\boldsymbol{w}}_T, S_T)$ and posterior predictive

$$p(y \mid \boldsymbol{x}, \mathcal{D}_T, \boldsymbol{\theta}) = \mathcal{N}(\bar{\boldsymbol{w}}_T^\top \boldsymbol{\phi}_{\boldsymbol{\theta}}(\boldsymbol{x}), \; \boldsymbol{\phi}_{\boldsymbol{\theta}}(\boldsymbol{x})^\top S_T \boldsymbol{\phi}_{\boldsymbol{\theta}}(\boldsymbol{x}) + \sigma^2) \quad . \tag{4}$$

Since the predictive distribution is Gaussian, it pairs straight-forwardly with conventional acquisition functions in Bayesian optimization and bandit tasks (Snoek et al., 2015; Riquelme et al., 2018).

Usually, such BLL models are trained using gradient descent on the exact (log) marginal likelihood over all data points. This can be computed either via the marginal likelihood or by backpropagating through the recursive last layer update (Harrison et al., 2018). The cost of this is prohibitive, as it requires iterating over the full dataset for each gradient computation. Moreover, it often is numerically unstable and can result in pathological behavior of the learned features (Thakur et al., 2020; Ober et al., 2021). An alternative strategy is to train on mini-batches (Snoek et al., 2015) to learn features and condition the last layer on the full dataset after training. However, this yields biased gradients and often results in a severely over-concentrated final model (Ober & Rasmussen, 2019).

To increase efficiency, recent work (Harrison et al., 2024; Watson et al., 2021) developed a deterministic variational lower bound to the exact marginal likelihood and proposed to optimize this instead, resulting in the *variational* Bayesian last layer (VBLL) model. In our setup, we fix a variational posterior $q(\boldsymbol{w}) = \mathcal{N}(\bar{\boldsymbol{w}}, S)$. The lack of subscript denotes that the last layer parameters belong to the variational posterior, and we will write $\boldsymbol{\eta} := (\bar{\boldsymbol{w}}, S)$ for convenience. Following Harrison et al. (2024, Theorem 1), the variational lower bound for regression with BLLs (under the prior defined previously) is

$$\log p(Y \mid X, \boldsymbol{\theta}) \geq \sum_t^T \left( \log \mathcal{N}(y_t \mid \bar{\boldsymbol{w}}^\top \boldsymbol{\phi}_t, \; \sigma^2) - \frac{1}{2}\boldsymbol{\phi}_t^\top S\boldsymbol{\phi}_t \sigma^{-2} \right) - \mathrm{KL}(q_{\boldsymbol{\eta}}(\boldsymbol{w}) \parallel p(\boldsymbol{w})) =: \mathcal{L}(\boldsymbol{\eta}, \boldsymbol{\theta}). \tag{5}$$

The variational posterior over the last layer is trained with the network weights $\boldsymbol{\theta}$ in a standard neural network training loop, yielding a lightweight Bayesian formulation. We refer to Appendix A for how VBLLs relate to other BNN methods we benchmark against in this work.

## 3 TRAINING AND MODEL SPECIFICATION

In this section, we discuss the training procedure we use for VBLLs in the BO loop. We first identify a relationship between variational training of Bayesian last layer models (as is used in the VBLL class of models) and recursive last layer computation (as is used in standard BLL models). We further present different methods for choosing whether to perform a recursive last layer update or full model retraining for a newly-observed datapoint. We then describe the feature training procedure, and discuss early stopping and previous model re-use. Finally, we present the overall training loop that combines these two training approaches, and relevant hyperparameters. We expand on all design decisions in the Appendix.

### 3.1 VARIATIONAL INFERENCE AND RECURSIVE LAST LAYER UPDATING

In this subsection, we describe a connection between the VBLL variational objective (5)—which is optimized through standard mini-batch optimization—and the recursive updating associated with Bayesian linear regression and BLL models.

**Theorem 1.** *Fix $\boldsymbol{\theta}$. Then, the variational posterior parameterized by*

$$(\bar{\boldsymbol{w}}^*, S^*) := \boldsymbol{\eta}^* = \arg\max_{\boldsymbol{\eta}} \mathcal{L}(\boldsymbol{\eta}, \boldsymbol{\theta}) \tag{6}$$

*is equivalent to the posterior computed by the recursive least squares inferential procedure described by* (2) *and* (3)*, iterated over the full dataset.*

This result follows from the fact that the true posterior for Bayesian linear regression is contained in our chosen variational family, and that the variational posterior is tight. The full proof is provided in Appendix B.1.

This equivalence provides a bridge between two different interpretations of model training: optimization-based training (as is used in NNs) and conditioning (as is used in GPs). This equivalence allows us to consider an online optimization procedure that interleaves two steps: full model training (including feature weights $\boldsymbol{\theta}$) on the variational objective, and last layer-only training via recursive conditioning. While the former optimizes all parameters and the latter only optimizes the last layer parameters, they are optimizing the same objective which stabilizes the interleaving of these steps.

We parameterize a dense precision (in contrast to standard VBLL models which parameterized the covariance) for the variational posterior by Cholesky decomposition, $S^{-1} = LL^\top$ (with $L$ lower triangular). The recursive precision update is then

$$L_t L_t^\top = \boldsymbol{\phi}_t \boldsymbol{\phi}_t^\top + L_{t-1} L_{t-1}^\top. \tag{7}$$

The updated Cholesky factor $L_t$ can be computed efficiently via a rank-1 update that preserves triangularity (Krause & Igel, 2015). The mean update can be computed via the natural parameterization

$$\boldsymbol{q}_t = \boldsymbol{\phi}_t y_t + \boldsymbol{q}_{t-1} \tag{8}$$

from which the last layer mean can be computed. We do not store the computation graph associated with these updates and do not backpropagate through model updates. This update has quadratic complexity in the dimensionality of $\phi$, and enables quadratic complexity for (efficient, randomized) computation of all terms in training. We discuss the numerical complexity of this procedure and the overall training loop in Appendix B.5.

## 3.2 FULL MODEL TRAINING

In full model training, we directly train the neural network weights $\boldsymbol{\theta}$, the variational posterior parameters $\boldsymbol{\eta}$, and the MAP estimate of the noise covariance $\sigma^2$ via gradient descent on (5).

**Training efficiency:** To improve training efficiency, we perform early stopping based on the training loss. Whereas standard neural network training will typically do early stopping based on validation loss, we directly terminate training when training loss does not improve for more than a set number of epochs. Whereas standard neural networks would typically substantially overfit with this procedure, we find that VBLL networks are not prone to severe overfitting. We additionally experiment with a more advanced form of continual learning in which we initialize network weights with previous training parameters. However, we find that the benefits from this are relatively minor and there are often exploration advantages to re-initializing the network weights at the start of training. Details of early stopping and model re-use are provided in Appendix B.2.

**Optimization:** For each iteration of model training, we will re-initialize the optimizer state. Because we are interleaving recursive updates, continued use of previous optimizer states such as momentum and scale normalization (as in e.g. Adam (Kingma & Ba, 2015)) actively harm performance. However, we do find benefits in using adaptive optimizers for training in general and use AdamW (Loshchilov & Hutter, 2017) throughout our experiments (with no regularization on the variational last layer). All training details are described in Appendix B.4.

## 3.3 CONTINUAL LEARNING TRAINING LOOP AND HYPERPARAMETERS

To alleviate VBLL surrogate fit times, we propose a continual learning training loop (Algorithm 1) which interleaves two training steps: full model training via minibatch gradient descent, and last layer conditioning.

**Choosing when to re-train model features:** There are several possible methods for choosing (for each new data point) whether to perform full model re-training or a last layer recursive update. We consider three methods in this paper. First, we consider a simple scheme of re-training the network every $M \geq 1$ steps, and otherwise doing recursive updates. Practically this performs poorly, and allocating compute to model training earlier in the episode performs substantially better. Therefore, we also consider a scheduling scheme in which re-training is randomized, with the probability of

---

**Algorithm 1** VBLL Bayesian Optimization Loop with Continual Variational Last Layer Training

---

**Require:** Model train frequency $T_{\text{model}}$ and total number of evaluations $T$; Wishart prior scale $V$
 1: $\mathcal{D} \leftarrow \mathcal{D}_{\text{init}}$ obtained from evaluations at a pseudo-random input locations from a Sobol sequence
 2: **for** $t = 0$ to $T$ **do**
 3:     $\gamma_{\text{reinit}} \leftarrow$ RE-INIT CRITERION$(\mathcal{D}, t)$ ▷ Periodic, scheduled, or event-triggered initialization
 4:     **if** $\gamma_{\text{reinit}}$ **then**
 5:         Init. model parameters $\boldsymbol{\eta}, \boldsymbol{\theta}, V$ and train via minibatch optimization on (5) using $\mathcal{D}$
 6:     **else**
 7:         Update $\boldsymbol{\eta}$ via (rank-1 Cholesky) recursive updates (2) and (3) using $(\boldsymbol{x}_{t-1}, y_{t-1})$ and $\Sigma$
 8:     **end if**
 9:     Select $\boldsymbol{x}_t$ via optimizing acquisition function; query objective function with $\boldsymbol{x}_t$ and receive $y_t$
10:     $\mathcal{D} \leftarrow \mathcal{D} \cup \{(\boldsymbol{x}_t, y_t)\}$
11: **end for**

---

re-training declining via a sigmoid curve over the course of the episode. Lastly, we consider a loss-triggered method for re-training, in which we re-train the full model if the log predictive likelihood of the new data under the model is below some threshold, and otherwise perform recursive updates. In the main body of the paper, we will only present the latter approach when performing continual learning. Details on re-initialization schemes and results for other scheduling methods are provided in Appendix B.3.

**Hyperparameters:** The VBLL models have several hyperparameters, some of which are similar to GPs and some of which are substantially different. We include hyperparameter studies in Appendix E. We specifically investigate the hyperparameter sensitivity of the noise covariance prior, network reinitialization rate and neural network architecture. Overall we find that the VBLL models are highly robust to these hyperparameters, but that proper tuning of them for the problem at hand can further improve performance.

## 4 ACQUISITION FUNCTIONS

In this section, we describe acquisition functions for our VBLL models in both the single objective and multi-objective case.

### 4.1 SINGLE OBJECTIVE ACQUISITION FUNCTIONS

VBLLs yield a Gaussian predictive distribution and thus most acquisition functions that are straightforward to compute for GPs are also straightforward for VBLLs. However, parametric VBLLs are also especially well suited for Thompson sampling compared to non-parametric models like GPs[2]. For Thompson sampling, we simply sample from the variational posterior of $\boldsymbol{w}$ at iteration $t$ and then construct a sample from the predictive $\hat{f}$ (cf. Fig. 1) as a generalized linear model as

$$\text{①} \quad \hat{\boldsymbol{w}} \sim q_{\boldsymbol{\eta}}^t(\boldsymbol{w}) \qquad \text{②} \quad \hat{f}(\boldsymbol{x}) := \hat{\boldsymbol{w}}^\top \boldsymbol{\phi}_{\boldsymbol{\theta}}(\boldsymbol{x}) \qquad \text{③} \quad \boldsymbol{x}_{t+1} = \arg\max_{\boldsymbol{x} \in \mathcal{X}} \hat{f}(\boldsymbol{x}) \quad (9)$$

This sample of the predictive can then be optimized *analytically*, which differs from classic Thompson sampling methods used for non-parametric GPs. This can similarly be done for LLLA BNNs (cf. Sec. 5), although Laplace approximated posterior samples are known to suffer from underfitting on training data (Lawrence, 2001; Roy et al., 2024). For the analytic optimization of the parametric sample, we use L-BFGS-B (Zhu et al., 1997) leveraging the fact that we can also easily obtain the gradient of $\hat{f}$. We initialize the optimization at multiple random initial conditions[3] and choose the best argmax as the next query location.

---

[2]Thompson sampling for GPs often involves drawing samples from high-dimensional posterior distributions generated at pseudo-random input locations (e.g., using Sobol sequences) and then selecting the argmax of the discrete samples as the next query locations (Eriksson et al., 2019). It is worth noting that while it is possible to construct analytic approximate posterior samples for GPs (Wilson et al., 2020; 2021), this approach is not yet commonly adopted in current practice, and is not possible to do for all kernels.

[3]In the following, the number of random initial conditions will be set to the same number of random initial conditions as for the standard optimization of the acquisition functions for a fair comparison.

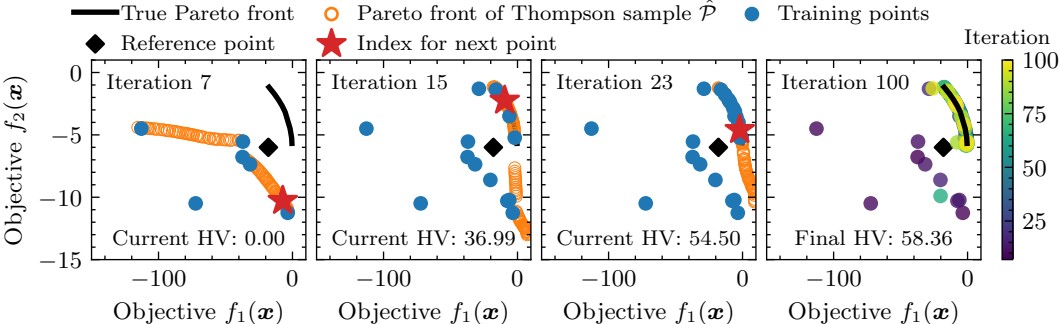

Figure 2: *Multi-objective Thompson sampling on BraninCurrin.* At each iteration, we optimize the multi-objective Thompson sample and choose as the next point the index that increases the predicted hypervolume the most. At the end of the optimization (right with colormap), we can observe that the true Pareto front is nicely approximated.

## 4.2 MULTI-OBJECTIVE ACQUISITION FUNCTIONS

In multi-objective BO it is common to model each objective with a separate GP (Zitzler et al., 2003; Daulton et al., 2020; Ament et al., 2024). While it is often preferable to model correlations between outputs (Swersky et al., 2013), this requires correlation kernels for the objectives, which may be difficult to specify. With neural network based surrogate models, one can simply set the number of outputs to the number of objectives, thus sharing feature learning between objectives. For VBLL networks, we can use the multivariate regression formulation to obtain a multivariate normal prediction for each point. Since this again is a Gaussian, it is straightforward to use popular acquisition functions for multi-objective optimization such as expected hypervolume improvement (Zitzler et al., 2003; Daulton et al., 2020; Ament et al., 2024).

Also for multi-objective optimization problems, we can leverage the parametric form of the VBLLs to do efficient Thompson sampling. It is also possible to do Thompson sampling with GPs for multi-objective optimization, but this usually involves approximations with a high number of Fourier bases functions (Bradford et al., 2018; Belakaria et al., 2019; Paria et al., 2020). For Thompson sampling with VBLLs, we first sample a NN realization of the VBLL network as in (9) and optimize the sample with NSGA-II (Deb et al., 2002) using Pymoo (Blank & Deb, 2020). This yields a predicted Pareto front $\hat{\mathcal{P}} := \{\hat{\boldsymbol{y}}_1, \ldots, \hat{\boldsymbol{y}}_P\}$ with size equal to the population size $P$ of the NSGA-II solver. In summary, the Thompson sampling procedure for multi-objective optimization is

$$\text{①} \quad \hat{W} \sim q_{\boldsymbol{\eta}}^t(W) \qquad \text{②} \quad \hat{\boldsymbol{f}}(\boldsymbol{x}) := \hat{W}^\top \boldsymbol{\phi}_{\boldsymbol{\theta}}(\boldsymbol{x}) \qquad \text{③} \quad \hat{\mathcal{P}} = \max_{\boldsymbol{x} \in \mathcal{X}} \hat{\boldsymbol{f}}(\boldsymbol{x}). \quad (10)$$

To choose the next candidate, we greedily select the index of the point in the predicted Pareto front that maximizes the improvement of the hypervolume. The hypervolume (HV), denoted by $\mathcal{HV}(\mathcal{P}, \boldsymbol{r})$, is a commonly used metric in multi-objective optimization, which measures the volume of the region dominated by the Pareto front $\mathcal{P}$ with respect to a predefined reference point $\boldsymbol{r}$ in the objective space. Our goal in multi-objective BO is to expand this dominated region by adding a new candidate to the Pareto front. At each iteration $k$, the current Pareto front is denoted by $\mathcal{P}_k$. To select the next candidate, we maximize the hypervolume improvement, which is the increase in hypervolume when adding a predicted new point $\hat{\boldsymbol{y}} \in \hat{\mathcal{P}}$ to the existing Pareto front $\mathcal{P}_k$ as

$$\text{④} \quad \boldsymbol{x}_{k+1} \in \arg\max_{\hat{\boldsymbol{y}} \in \hat{\mathcal{P}}} \mathcal{HV}(\mathcal{P}_k \cup \{\hat{\boldsymbol{y}}\}, \boldsymbol{r}). \quad (11)$$

If no points in the predicted Pareto front improve the hypervolume or the solution is non-unique, we randomly sample from the set of maximizers. Figure 2 shows Thompson sampling with VBLLs on the BraninCurrin benchmark (cf. Sec. 5). After the initialization, none of the points in the Pareto front $\hat{\mathcal{P}}$ improve the HV. Therefore a random index from the list of solutions in $\hat{\mathcal{P}}$ is chosen for the next query location. After some iteration, the $\hat{\mathcal{P}}$ partially includes the true Pareto front. The algorithm continuously samples points that improve the HV until after 100 iterations $\mathcal{P}$ is well approximated.

In all experiments in this paper, we assume the reference point to be known but we should note that it is also possible to estimate or adaptively choose the reference point $r$, which can improve the effectiveness of the optimization process (Bradford et al., 2018). Additionally, this strategy of maximizing hypervolume can be combined with other acquisition functions; of particular interest are information-theoretic acquisition functions as in the single objective case. For instance, max-value entropy search for multi-objective optimization (Belakaria et al., 2019) relies on random Fourier features to approximate samples from the GP posterior; this can easily be switched to using Thompson sampling with VBLL networks.

## 5    RESULTS

We evaluate the performance of the VBLL surrogate model on various standard benchmarks and three more complex optimization problems, where the optimization landscape is non-stationary. For experimental details and ablations of hyperparameters, we refer the reader to Appendix D and E.

### 5.1    SURROGATE MODELS

We use the following baselines throughout the experiments in addition to **VBLLs**: Gaussian Processes (**GPs**), Infinite-Width Bayesian Neural Networks (**I-BNNs**) (Lee et al., 2018; Li et al., 2024), Deep Kernel Learning (**DKL**) (Wilson et al., 2016), Last Layer Laplace Approximations (**LLLA**) (Daxberger et al., 2021; Kristiadi et al.), and Deep Ensembles (**DE**). For the configuration of neural network backbone of the BNN baselines as well as the VBLLs, we closely follow the setup in Li et al. (2024). For details on all baselines and their configurations, refer to Appendix D.1.

In all subsequent experiments, we select the number of initial points for the single objective benchmarks equal to the input dimensionality $D$ and for the multi-objective benchmarks we use $2 \cdot (D+1)$ initial points (Daulton et al., 2020; Balandat et al., 2020). We further use a batch size of one as in the classic BO setting.[4] We compare the performance of all surrogates for the following acquisition functions: *(i)* log expected improvement (`logEI`) (Ament et al., 2024), a numerically more stable version of standard expected improvement, *(ii)* Thompson sampling (`TS`) (Thompson, 1933; Russo et al., 2018), *(iii)* and log expected hypervolume improvement `logEHVI` (Ament et al., 2024) for the multi-objective benchmarks.

### 5.2    BENCHMARK PROBLEMS

We begin by examining a set of standard benchmark problems commonly used to assess the performance of BO algorithms (Eriksson et al., 2019; Ament et al., 2024). Figure 3 (top) illustrates the performance of all surrogates on these benchmark problems. It can be observed that, as expected, GPs perform well. The BNN baselines also demonstrate strong performance on lower-dimensional problems, although they do not match the performance of GPs on the `Hartmann` function. VBLLs, however, generally perform on par or better than most other BNN surrogates on the classic benchmarks. Interestingly, for `TS`, we notice that on the `Ackley2D` and `Ackley5D` benchmark, the VBLLs with analytic optimization of the Thompson samples even surpass the performance of GPs as well as all other baselines. The continual learning baselines show slightly worse performance than the standard VBLLs but at the benefit of reduced computing.

### 5.3    HIGH-DIMENSIONAL AND NON-STATIONARY PROBLEMS

GPs without tailored kernels often struggle in high-dimensional and non-stationary environments (Li et al., 2024); areas where deep learning approaches are expected to excel. Our results on the $200D$ `NNdraw` benchmark (Li et al., 2024), the real-world $25D$ `Pestcontrol` benchmark (Oh et al., 2019), and the $12D$ `Lunarlander` benchmark (Eriksson et al., 2019) are shown in Figure 3 (bottom). On these benchmarks, all BNN surrogates show strong performance; especially LLLA, DEs and the VBLL networks. While GPs perform well with `logEI` on `NNdraw` and the I-BNNs show good performance on `Pestcontrol`, the VBLLs and LLLA models are consistently the best-performing surrogates for `TS`. Similar to the classic benchmarks, the continual learning version of the VBLLs

---

[4]Note that this differs from the experimental setup by Li et al. (2024) which use larger batch sizes.

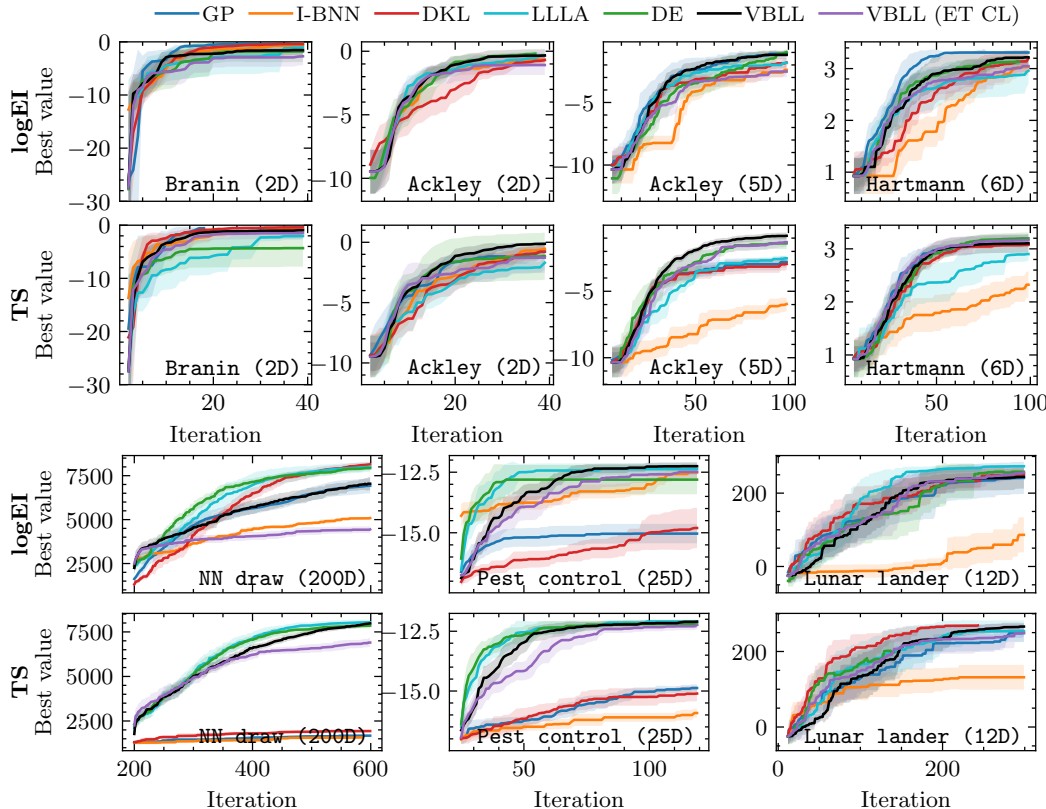

Figure 3: *Classic benchmarks (top) and high-dimensional and non-stationary benchmarks (bottom).*
Performance of all surrogates for `logEI` (top) and `TS` (bottom).

shows similar performance to the VBLLs. Additionally, we note that by adjusting the Wishart scale or changing the re-initialization strategy we can further enhance the performance of continual learning for `logEI` (cf. Appendix E.2 and Appendix B.3.3, respectively).

## 5.4 IMPACT OF RE-INITIALIZATION METHODS

We compare our two strategies for determining when to re-initialize the model: event-triggered and scheduled re-initialization (cf. Appendix B.3). For this, we define the indicator function $\mathbb{I}(t)$, which is 1 if the model is re-initialized at iteration $t$ and 0 if updated recursively. We show the expected value of this indicator function as well as the performance (top) over ten seeds (bottom) in Fig. 4. We can observe that both strategies perform similarly for `TS`, achieving near-VBLL performance with large reductions in runtime as highlighted by the percentage decreases. Further, the event trigger adapts more flexibly than the pre-defined sigmoid schedule, particularly for Branin, where it reduces runtime early on. In Appendix B.3.3, we included a longer discussion on the comparison of the re-initialization strategies. We find

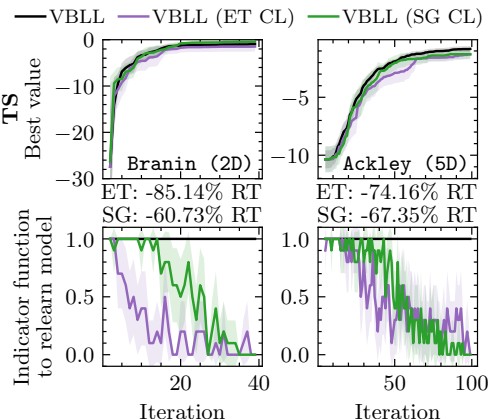

Figure 4: Impact of event-triggered (ET) and sigmoid scheduled (SG) re-initialization on the performance and runtime (RT) saving.

that the event-triggered approach provides greater adaptivity compared to the sigmoid schedule but can lead to stagnation in high-dimensional settings, as observed for NNdraw with `logEI` in Fig. 3.

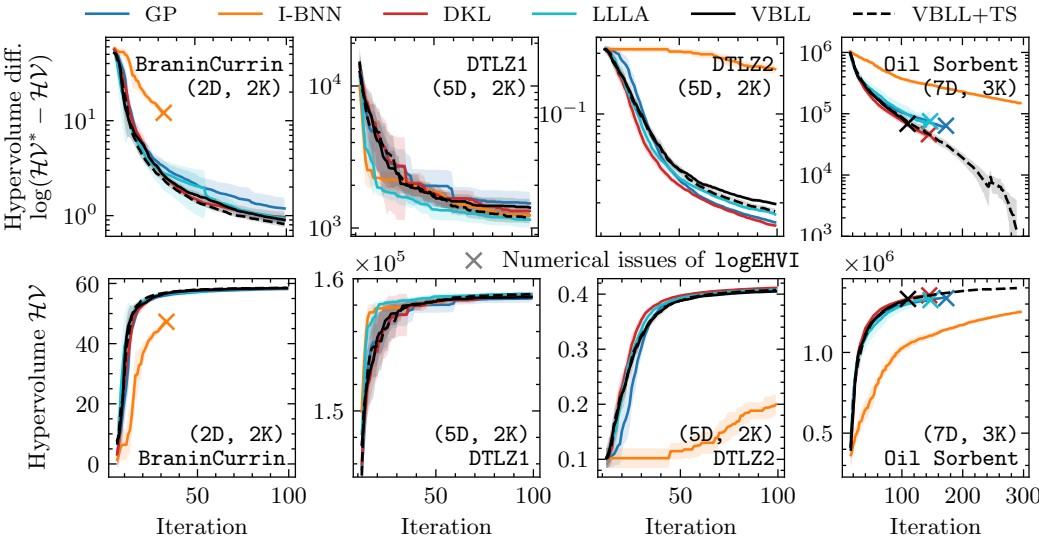

Figure 5: *Multi-objective benchmarks.* Performance of all surrogate models using `logEHVI` and VBLLs with `TS`. A cross indicates the crash of a surrogate's furthest seed due to a numerically unstable acquisition function. VBLL+TS successfully navigates the numerically unstable HV plateau in OilSorbent, enabling a more accurate approximation of its three-dimensional Pareto front.

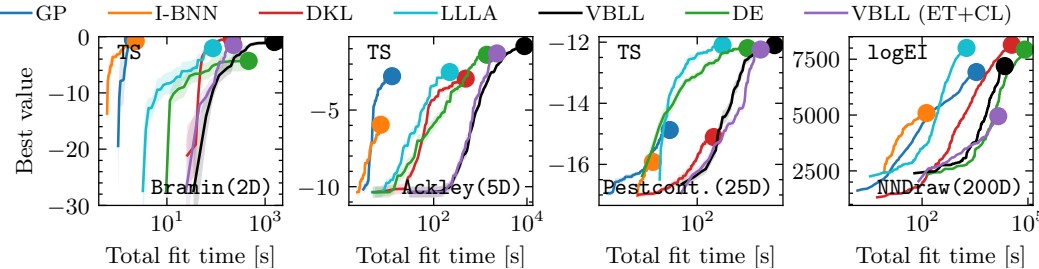

Figure 6: *Performance vs. accumulated surrogate fit time.* VBLLs are the most expensive to train surrogate model. Using the proposed continual learning scheme can significantly reduce runtime while maintaining good performance.

Interestingly, this stagnation does not occur as early when using `TS`, suggesting that the additional randomness helps mitigate this issue.

## 5.5 MULTI-OBJECTIVE PROBLEMS

We evaluate our approach on multi-objective optimization problems. Here, we consider the standard benchmarks BraninCurrin ($D = 2$, $K = 2$), DTLZ1 ($D = 5$, $K = 2$), DTLZ2 ($D = 5$, $K = 2$), and the real world benchmark Oil Sorbent ($D = 7$, $K = 3$) (Wang et al., 2020; Li et al., 2024). Figure 5 shows the performance of all surrogate in terms of the obtained HV with respect to fixed reference points, as well as the logarithmic HV difference between the maximum HV[5] and the obtained HV as common metrics in multi-objective BO (Belakaria et al., 2019; Daulton et al., 2020). The performance of all BNN models is similar. It is however notable that on OilSorbent all baselines using `logEHVI` crash due to numerical issues while optimizing the acquisition function. Such numerical issues are known for expected improvement type acquisition functions due to vanishing gradients (Ament et al., 2024). Thompson sampling with VBLLs does not result in numerical issues, and we observe that with this combination of model and acquisition function, the Pareto front can be further refined.

---

[5]For BraninCurrin, DTLZ1, and DTLZ2 we use the values provided by BoTorch, and for OilSorbent we estimate the maximum HV based on the asymptotic performance of the best performing surrogate.

## 5.6 TRAINING TIME COMPARISON

Lastly, we consider a surrogate fit time comparison in which we keep constant BO budget and track accumulated fit time to highlight that recursive updates can significantly speed up run time. It should be noted that in the usual BO setting, we assume that the black-box function is expensive to evaluate, and hence, fit times play only a minor role in many applications, such as drug- or materials discovery. Still, there may be applications where fit times are important. In Figure 6, we observe that VBLLs are the most expensive surrogates to train on smaller problems, while on NNDraw, DKL and DE are the most expensive. The event-triggered re-initializing combined with recursive updates of the variational posterior significantly reduces the runtime while maintaining good performance.

## 6 DISCUSSION AND CONCLUSIONS

In this paper, we have developed a bridge between classical conjugate Bayesian methods for BO and BNN-based methods. We built upon VBLLs (Harrison et al., 2024) to obtain the expressivity of neural networks, together with the simple and robust uncertainty characterization associated with GPs.

**Continual learning:** We introduced an online training scheme that enables online complexity comparable to GPs. While we have developed an efficient training scheme, further developments in combining continual learning with our architecture are possible. For our experiments, we use a simple threshold on the log-likelihood to decide on re-initializing the model. Here, further concepts from model selection can likely be used to make even more informed decisions. While our approach was effective, many methods in continual learning (e.g. Nguyen et al. (2018)) have been developed to improve training efficiency. While the combination of previously-developed continual learning schemes with our last layer updating may be non-trivial, this is likely a promising direction for improving efficiency. Balancing the functional randomness afforded by network re-initialization with efficient continual learning will likely be a necessary focus of future work in this direction.

Additionally, as VBLLs are a new model class, we believe that there exist various options beyond the approaches developed herein that can be used to speed up runtime further. In our experiments, we used the default initialization provided in Harrison et al. (2024). Further investigating the initialization of the VBLLs through a form of stochastic feature re-use and its effect on fit time can likely improve wall clock performance. Also, standard deep learning best practices, such as learning rate scheduling, could be beneficial, and exploring these directions is promising if wall clock time is important.

**Performance and baselines:** VBLLs perform on par with GPs on standard low-dimensional benchmarks, yet significantly outperform GPs in high-dimensional and non-stationary problems. Furthermore, VBLLs outperform a wide array of Bayesian neural network models, including I-BNNs (Adlam et al., 2021), Laplace approximation-based models (Daxberger et al., 2021), deep ensembles (Hansen & Salamon, 1990; Lakshminarayanan et al., 2017) and deep kernel learning (Wilson et al., 2016). Especially the combination of VBLLs with TS shows strong performance throughout. While Laplace-approximation methods perform well in our experiments, we find that they appear more sensitive to observation noise (see Appendix C.3 for experimental results highlighting this).

**Acquisition functions:** Because of the parametric architecture of VBLLs, they can efficiently be combined with Thompson sampling as an acquisition function. This yields reasonable benefits in the univariate case (as shown by our full set of results) but yields substantial benefits in the multi-objective case due to better numerical conditioning. Moreover, the ease of use with Thompson sampling also enable easy use of TS-based information-theoretic acquisition functions such as max-value entropy search (Wang & Jegelka, 2017; Belakaria et al., 2019).

## ACKNOWLEDGMENTS

The authors thank Friedrich Solowjow, Alexander von Rohr, Kevin Murphy, and Zi Wang for their helpful comments and discussions. Paul Brunzema is partially funded by the Deutsche Forschungsgemeinschaft (DFG, German Research Foundation)–RTG 2236/2 (UnRAVeL). Simulations were performed in part with computing resources granted by RWTH Aachen University under projects rwth1579 and p0022034.

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

## A    RELATED WORK COMPARISONS

### A.1    VBLL COMPARISON TO LAST LAYER LAPLACE APPROXIMATIONS

Laplace methods approximate the predictive covariance with the (inverse) Hessian. In the line of work on Laplace approximations for neural networks (e.g. Daxberger et al. (2021)), the Fisher information matrix—which consists of the outer-product of network gradients, summed over all data—is used as an approximation to the Hessian. For last layer Laplace in particular, this reduces to the outer product of features, equivalent to covariance computation for Bayesian last layer methods. The primary difference is Daxberger et al. (2021) also typically optimize the hyperparameters (including the last layer prior and hyperparameters) via an empirical Bayes objective. Thus, the last layer Laplace method in this setting can be interpreted as a form of the (non-Variational) Bayesian last layer model. While we optimize the noise covariance we do not optimize the last layer prior, which is the primary difference between our VBLL-based approach and last layer Laplace approximations. Another key difference is that in the presented VBLL models, the variational posterior is learned *jointly* with the features (or basis functions), which is not the case for last layer Laplace where the approximate posterior distribution is learned post-hoc fitting of the features.

### A.2    VBLL COMPARISON TO DEEP KERNEL LEARNING

Deep kernel learning (DKL) (Wilson et al., 2016) aims to learn an input transform to a kernel with a deep neural network to effectively capture non-stationary and complex correlations in the input space. Specifically, it uses a neural network $g_\theta$ as input wrapping to the kernel as $k_{\mathrm{DKL}} := k(g_\theta(\boldsymbol{x}), g_\theta(\boldsymbol{x}'))$. The kernel's hyperparameters and the neural network are trained simultaneously by optimizing the exact (log) marginal likelihood of the GP posterior on the full data set. With the learned kernel, inference is identical to standard non-parametric GPs through explicit conditioning on past data and therefore scales as $\mathcal{O}(N^3)$. This explicit conditioning can become especially problematic for larger datasets. Here, parametric models such as VBLLs and LLLA, which are trained on mini-batches, are preferable. Intuitively, VBLLs can be considered a parametric version of DKL, also learning correlations by optimizing the variational posterior of the last layer in tandem with the feature-extracting backbone network. Optimizing DKL is challenging primarily because of the computational complexity of gradient computation with a marginal likelihood objective and overparameterized models (Ober et al., 2021). However, it can still yield good performance on many benchmarks which we also demonstrate in Figure 3. Still, similar to GPs, it struggles in high-dimensional Thompson sampling due to its non-parametric nature.

## B    ALGORITHMIC DETAILS

### B.1    PROOF OF THEOREM 1

We restate the theorem for completeness.

**Theorem 1.** *Fix $\boldsymbol{\theta}$. Then, the variational posterior parameterized by*

$$(\bar{\boldsymbol{w}}^*, S^*) := \boldsymbol{\eta}^* = \arg\max_{\boldsymbol{\eta}} \mathcal{L}(\boldsymbol{\eta}, \boldsymbol{\theta}) \tag{12}$$

*is equivalent to the posterior computed by the recursive least squares inferential procedure described by* (2) *and* (3)*, iterated over the full dataset.*

*Proof.* There are two methods to prove this result. First, we can note that for the standard ELBO—constructed from a single application of Jensen's inequality, exchanging the log and the expectation—the variational lower bound is tight if the chosen variational family contains the true posterior (see e.g. Knoblauch et al. (2019) for discussion). Our chosen variational posterior formulation consists of independent posteriors for each output dimension, with dense covariances. This contains the exact posterior for multivariate Bayesian linear regression with diagonal $\Sigma$ and thus the variational lower bound is tight under our modeling assumptions. Moreover, because the variational lower bound (for fixed $\boldsymbol{\theta}$) is strictly concave for appropriate choice of priors, the maximizer is unique.

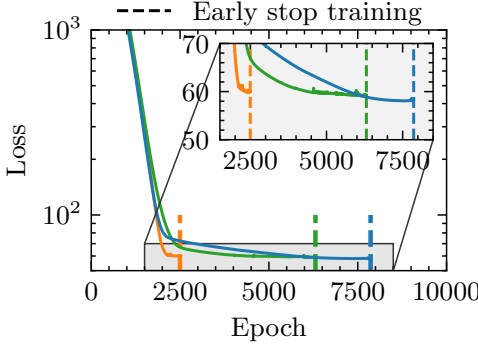
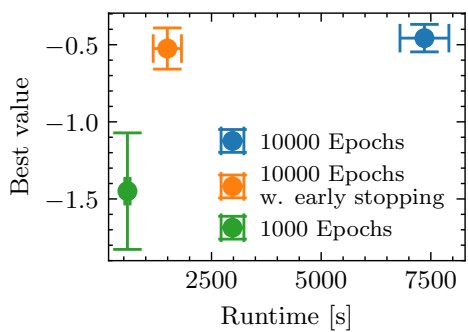

(a) Training loss with early stopping for three different model initializations.

(b) Comparison of runtime and performance for different fixed epochs and using patience on Ackley5D.

Figure 7: Illustration of early stopping in VBLL training (left) and its impact on runtime and performance (right). By introducing the early stopping, we can significantly reduce runtime while maintaining good performance.

A second proof approach is constructive: note that the variational lower bound is strictly concave and compute the maximizer by computing the gradient and using first order necessary conditions for optimality. By inspection one can see that this is equivalent to the recursive least squares posterior. ☐

### B.2 MODEL RE-USE AND EARLY STOPPING

**Early stopping:** The training process for BNNs typically follows standard neural network procedures and relies on a held-out validation set to determine when to stop training (Watson et al., 2021; Harrison et al., 2024). However, this approach is impractical in the context of BO with limited data, particularly in the early stages of optimization when the training data size is very small. To address this, we implement early stopping for VBLL networks based on the training loss. Specifically, we track the average loss for each training epoch and stop if the average loss does not improve for $M$ consecutive epochs, using the model parameters that produced the lowest training loss.

Figure 7a shows the training of the same VBLL network with different initialization but using identical data and optimizer settings. The dashed vertical lines indicate that the training loss reaches a plateau at significantly different epochs across these initializations. Such a large variance in the optimal number of training epochs was also observed in the original VBLL paper for regression tasks (Harrison et al., 2024). By applying an early stopping criterion, we are able to specify a high maximum number of epochs while still reducing computational costs, as each initialization is only trained *as long as necessary* to achieve good empirical performance. In Figure 7b, we can see that by using early stopping, the runtime can be reduced while maintaining performance. Only training for a small fixed number of epochs has similar savings in runtime but at the cost of final performance.

It is important to emphasize that this approach is not exclusive to VBLLs; we believe that applying a similar early stopping criterion could prove beneficial for training neural network-based surrogate models in BO more broadly. In our experiments, we therefore applied early stopping to all applicable BNN baselines.

**Feature re-use:** The naive approach to training requires training a new network for each step of full model training. While we mitigate the cost of training via the last layer recursive update, we further explore using continual feature learning for faster convergence. For this, we initialize the VBLLs at each (full model training) iteration with the feature weights from the last full model training iteration, and the variational posterior from the previous recursive update iteration. This warm start, combined with early stopping, yields substantial speed-ups in training. This feature re-use approach follows conventional continual learning, in which networks are trained for streaming data (Zenke et al., 2017; Hadsell et al., 2020). However, as discussed in Appendix C.1, re-initializing networks plays an important role that has strong connections to Thompson sampling over the features, and thus we will interleave continual (feature) training with occasional model re-initialization.

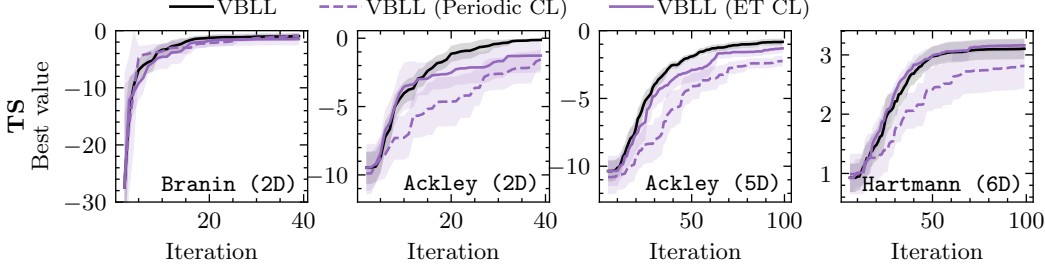

Figure 8: Comparison of periodic retraining (Periodic CL) and event-triggered retraining (ET CL).

## B.3 RECURSIVE UPDATE SCHEDULING

We next discuss the different approaches for deciding when to re-initialize the model. There are different heuristics one can apply to decide when to re-initialize the network. A straightforward approach would be to fix a re-initialization rate $M \geq 1$ a-priori.

### B.3.1 MODEL RE-INITIALIZATION THROUGH EVENT TRIGGERING

A simple rule for deciding online whether to re-initialize the network is to check how consistent a new observation is with the current variational posterior. For this, we simply compute the log-likelihood of the incoming point and compare it against a threshold $\Lambda$ as

$$\log \mathcal{N}_{q_\eta^t}(y_t \mid \boldsymbol{x}_t) < \Lambda \iff \text{re-initialize model} \tag{13}$$

Here, $\Lambda$ is a user-defined threshold and $\mathcal{N}_{q_\eta^t}$ is the predictive under the current variation posterior. In all our experiments, we set $\Lambda = 0$ and did not further tune this. Intuitively, this gives practitioners a tuning nob trading off between computational efficiency and predictive accuracy. In Figure 9, we show the influence of $\Lambda$ on both final performance and runtime.

In Figure 8, we compare the event-triggered strategy to periodic retraining ($M = 5$) demonstrating an improved performance on all benchmarks. However, it should be noted that with periodic retraining, the total runtime can be better estimated as a fraction of retraining at every time step, whereas this is not possible with the event-triggered strategy.

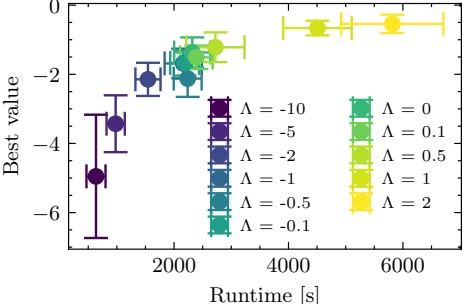

### B.3.2 MODEL RE-INITIALIZATION THROUGH SCHEDULING

Another idea to decide whether to re-initialize the model or perform a recursive update is through scheduling. Various scheduling approaches can be leveraged here, but scheduling re-initialization with a sigmoid arises naturally: in the beginning, when data is still spare, we want a high retrain rate, whereas in

Figure 9: Performance and runtime for various thresholds $\Lambda$ on Ackley5D with TS. The threshold defines a trade-off between final performance and total runtime.

the later stages, the basis functions of the model are expressive enough to model the function. We can then use efficient recursive updates. Specifically, we parameterize the probability of re-training with a sigmoid taking as input the current time step

$$\text{①} \quad p(t) = \frac{1}{1 + e^{-s(c-t)}} \qquad \text{②} \quad Z_t \sim \text{Bernoulli}(p(t)) \iff \text{re-initialize model.} \tag{14}$$

Here, $c$ is the center of the sigmoid and $s$ is the stretch parameter. We set the stretch parameter as $s = \frac{2\ln(9)}{T \cdot w}$ where $w$ is the transition window ratio.

**Remark.** *Setting the stretch parameter $s = \frac{2\ln(9)}{T \cdot w}$ ensures that the sigmoid transition from $p(t) = 0.9$ to $p(t) = 0.1$ occurs over $w \cdot T$ time steps. With the sigmoid as $p(t) = \frac{1}{1+e^{-s(c-t)}}$ we can set*

$p(t_1) = 0.9$ *and* $p(t_2) = 0.1$, *and solve for* $t_1$ *and* $t_2$ *as* $t_1 = c - \frac{\ln(9)}{s}$ *and* $t_2 = c + \frac{\ln(9)}{s}$. *The length of the transition window is* $t_2 - t_1 = \frac{2\ln(9)}{s}$. *Setting this equal to the desired window* $w \cdot T$, *we have* $\frac{2\ln(9)}{s} = w \cdot T$ *which implies* $s = \frac{2\ln(9)}{T \cdot w}$.

As defaults, we set the center to $T/2$ and $w = 0.5$. However, similar to the threshold for the event trigger, practitioners can use these tuning knobs to prioritize computational efficiency or predictive accuracy.

### B.3.3 COMPARISON OF RE-INITIALIZATION STRATEGIES

We next compare the two main strategies for determining when to re-initialize the model. For this, we define the following indicator function:

$$\mathbb{I}(t) = \begin{cases} 1 & \text{if model is re-initialized at iteration } t, \\ 0 & \text{if model is updated recursively at iteration } t. \end{cases} \tag{15}$$

We compare the performance and re-initialization behavior of the approaches on a selected subset of experiments in Figure 10. We can observe that for TS, both approaches perform similarly and almost

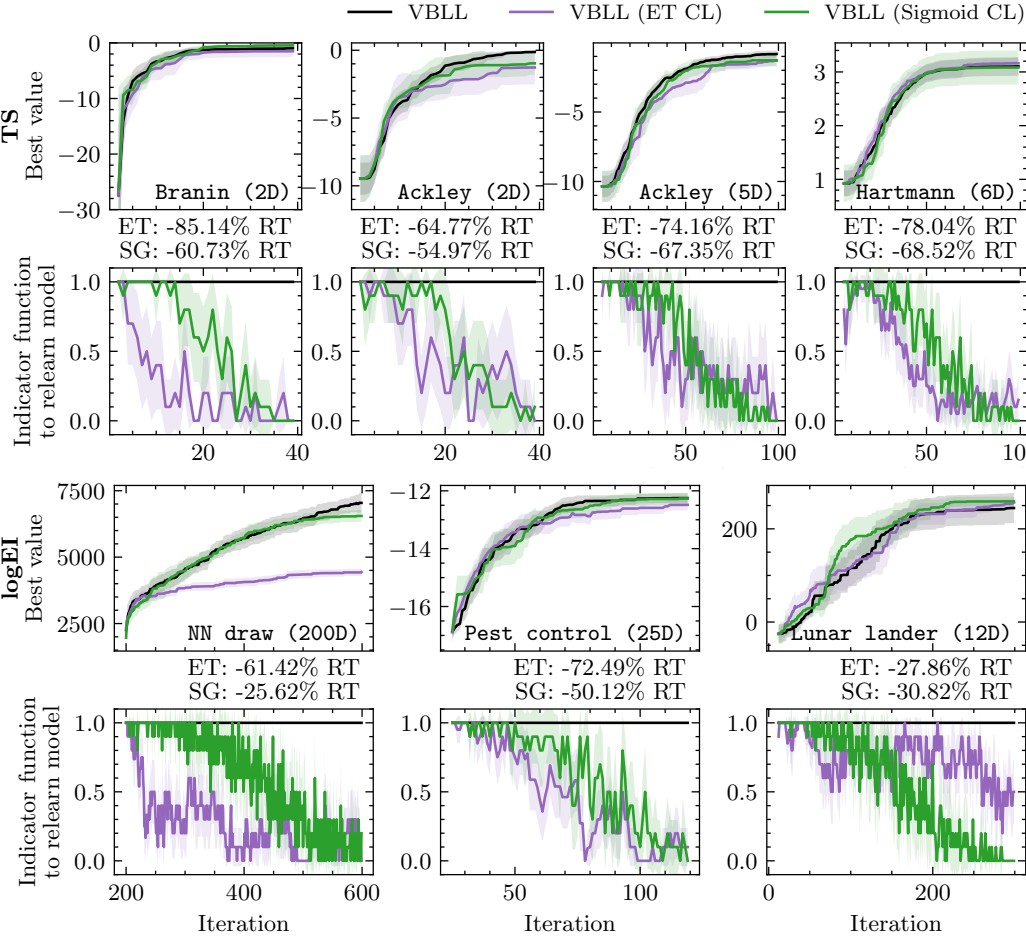

Figure 10: Comparison of event-triggered and scheduled re-initialization on a selected subset of problems. The top row always shows the performance, and the bottom row shows $\mathbb{E}[\mathbb{I}(t)]$ across seeds. In between, we report the improvement in runtime compared to the standard VBLL baseline.

on par with the VBLLs but at a significant reduction in total runtime. Through all experiments, we can observe the more adaptive behavior of the event trigger compared to the pre-defined sigmoid schedule. Specifically for Branin, we can observe that the event trigger can significantly reduce the runtime already early on.

For high-dimensional problems using `logEI`, we observe a notable difference in performance for NNdraw. When employing the event trigger, the algorithm stagnates resulting in poor performance. In contrast, sigmoid scheduling allows the algorithm to achieve performance levels that approximate those of the VBLLs. Interestingly, we do not observe the same behavior when using `TS` (cf. Figure 3). Here, the additional exploration introduced by Thompson sampling appears to mitigate these issues. Based on these observations, we hypothesize that especially the interplay of `TS` with an online strategy leveraging event-triggered re-learning offers significant potential for practical applications.

Finally, it is important to acknowledge that the comparisons in this section are based on a single set of parameters, limiting the scope for direct comparisons, particularly due to differences in runtime savings. Moreover, while both strategies arise naturally, an intriguing direction for future work would be to investigate more advanced approaches, such as those from model selection.

## B.4 TRAINING DETAILS

For training the VBLL models, we closely follow Harrison et al. (2024). For all experiments, we use AdamW (Loshchilov & Hutter, 2017) as our optimizer with a learning rate of $10^{-3}$, set the weight decay for the backbone (*not* including the parameters of the VBLL) to $10^{-4}$, and use norm-based gradient clipping with a value of $1$. For the VBLL, we set the prior scale to $1$ and the the Wishart scale to $0.01$. A sensitivity analysis of the Wishart scale on the performance and run time is in Appendix E. As mentioned in Sec. 3, we employ early stopping for all VBLL models based on the training loss. We track the average loss of a training epoch and if this average loss does not improve for a $100$ epochs in a row, we stop training and use the model parameters that yielded the lowest training loss.

## B.5 COMPUTATIONAL COMPLEXITY

The update formula for the last layer (defined by the rank-1 Cholesky update for the covariance and (8)) has quadratic computational complexity in the feature dimension. Computing $\bar{w}_t$ can similarly be computed with quadratic complexity as the product $L_t^{-\top} L_t^{-1} q_t$. This can be computed via efficient linear solves due to the triangularity of $L_t$. We also note that computing the log determinant $-\log \det S_t$ is equivalent to computing $\log \det S_t^{-1} = 2 \log \det L_t$, which is equal to the sum of the log diagonal elements, and thus has linear complexity.

The KL penalty on the last layer variational posterior requires computing the trace of the covariance, $\mathrm{tr}(L_t^{-1} L_t^{-\top})$. This can be expressed as $\|L_t^{-1}\|_F^2$, and complexity for this is cubic. We can turn instead to randomized algorithms—in particular, Hutchinson estimators (Hutchinson, 1989)—to yield an approximate solution with quadratic complexity. This estimator is

$$\mathrm{tr}(L_t^{-\top} L_t^{-1}) = \mathbb{E}[\mathbf{1}^\top \mathrm{solve}(L_t, \epsilon)^2] \approx \frac{1}{N} \sum_{i=1}^{N} \mathbf{1}^\top \mathrm{solve}(L_t^\top, \epsilon_i)^2 \qquad (16)$$

for appropriately sampled $\epsilon$ (for example, $\epsilon \sim \mathcal{N}(0, I)$). Here, $\mathrm{solve}(\cdot, \cdot)$ denotes a linear solve. Thus this estimator is quadratic in the feature dimension and linear in $N$. The MSE convergence rate for this estimator is linear, but modifications exist that achieve quadratic convergence (Meyer et al., 2021) by exploiting quasi-Monte Carlo methods.

With the exception of the trace computation, the computational complexity is quadratic in the feature dimension. Therefore, for an exact solution the complexity is cubic (due to the cubic complexity trace computation), and a rapidly convergent randomized solution is possible with quadratic complexity.

## B.6 FURTHER DETAILS ON THE SETUP AND ACQUISITION FUNCTIONS

We implement the VBLLs within BoTorch (Balandat et al., 2020). We further build on the implementation of Li et al. (2024) for the different baselines which are also based on BoTorch as well as GPyTorch (Gardner et al., 2018). As best-practice in BO, we standardize the data to mean zero and a variance of one at each iteration. We further transform the input space specified by the problem intro the hypercube $\mathcal{X} \in [0, 1]^D$. For completeness, we again list all the baselines below and then discuss the optimization of the acquisition functions.

**Acquisition functions:** We use 10 restarts and 512 raw samples for optimizing the acquisition functions `UCB` and `logEI` for all models. For `TS` we optimize the analytic sample of the VBLLs

with 10 random restart using L-BFGS-B (Zhu et al., 1997) as the optimization method. For `TS` from the non-parametric models, we use the same heuristic as in Eriksson et al. (2019) and generate $\min\{5000, \max\{2000, 200 \cdot D\}\}$ pseudo-random input points from a Sobol sequence and then sample from the high-dimensional multi-variate normal. The next location is then the argmax of the sample path. For the multi-objective problems, we use `logEHVI` for all models with consistent reference points. For `TS` with VBLLs, we follow the procedure described in Sec. 4. For NSGA-II, we use a population size of 100 and terminate after 200 generations.

## C  DISCUSSION AND BASELINE ABLATIONS

### C.1  ON NEURAL NETWORK THOMPSON SAMPLING

We further investigated the use of using neural network based Thompson samples. With VBLLs, we effectively maintain a distribution over plausible deterministic NNs that are in accordance with the current data set and noise level. Instead of maintaining such a distribution and then sampling from the variational posterior of the weights $w$, a straight-forward idea would be to *directly* train a deterministic neural network using the same optimizer (including the same early stopping etc.), and L2 loss, as well as L2 regularization for the backbone directly generating a MAP Thompson sample. Note that with such an approach, one would no longer be able to leverage acquisition functions that rely on good uncertainty quantification such as `logEI` or more sophisticated such as information-theoretic acquisition functions based on entropy search such as MES Wang & Jegelka (2017), which may require large ensembles (up to 100 NNs), making it computationally expensive. Still, we tested this baseline and the results are summarized in Fig. 11.

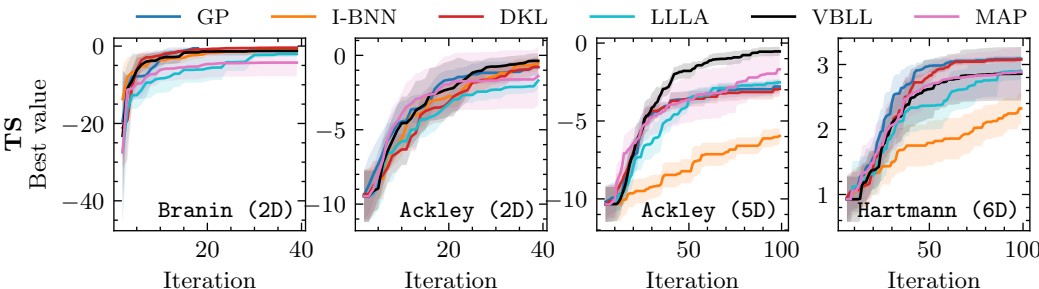

Figure 11: *Comparison of neural network Thompson sampling on synthetic benchmarks.* The deterministic MAP Thompson sample shows surprisingly good performance however also yields large variance on simple benchmarks which is undesirable.

Here `MAP` refers to the MAP Thompson sample baseline. We can observe that this simple baseline performs surprisingly well but cannot match the performance of the other baselines. We also observe that in relatively simple problems, such as `Branin` and `Ackley2D`, the variance is significantly larger compared to the other baselines, which is undesirable in BO applications. We also compared this baseline on the high-dimensional and real-world benchmarks and the results are shown in Fig. 12.

The MAP approach demonstrates superior performance on the `NNdraw` task. This is likely because the Wishart scale in the VBLL baselines is not well-tuned for the `NNdraw` problem (cf. Sec. E). The data is normalized to zero mean and a standard deviation of one; however, with a Wishart scale greater than zero, the assumed noise in this normalized space affects the accuracy of the correlations between data points–especially for the large value range in `NNdraw`. In contrast, the MAP baseline does not, by design, account for noise, which may contribute to its better performance in this context. Additionally, the MAP baseline exhibits slightly faster convergence on `Pestcontrol`. For `Lunarlander`, the MAP approach shows considerable variance and fails to match the final performance of the VBLLs.

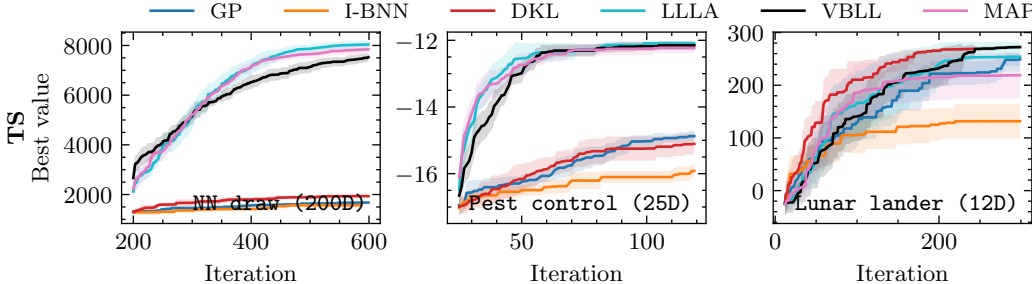

Figure 12: *Comparison of neural network Thompson sampling on high-dimensional and real-world benchmarks.* On these benchmarks, the MAP Thompson sample baseline shows mixed performance. It performs better on `NNdraw` but exibits large variance for `Lunarlander`.

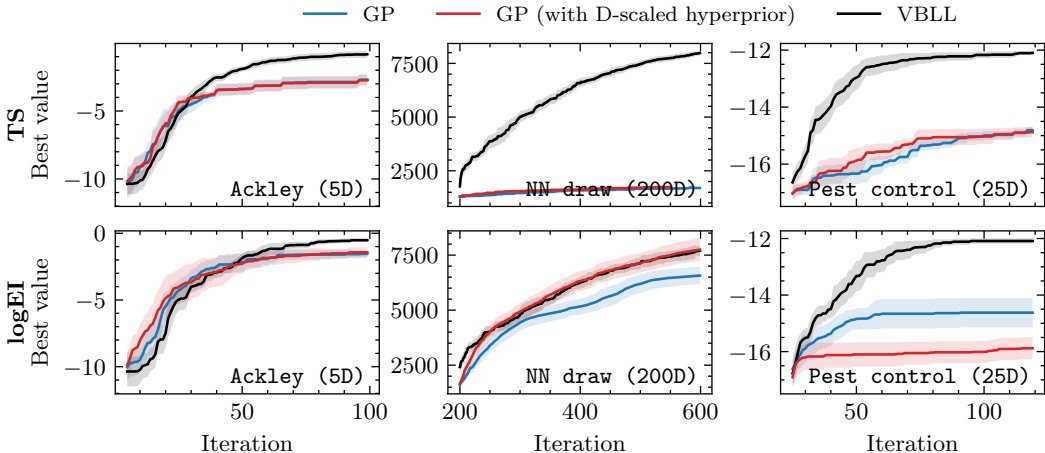

Figure 13: Comparison of a standard GP with and without a $D$-scaled hyperprior to VBLLs.

## C.2 COMPARISON TO $D$-SCALED GAUSSIAN PROCESS PRIORS

We further compare the VBLLs to a Gaussian process with so-called $D$-scaled hyperpriors on all the lengthscales which has demonstrated promising results in high-dimensional settings (Hvarfner et al., 2024). With this, we aim to test the hypothesis that the VBLL's improvement in performance over the GPs on tasks such as Pestcontrol does not come from bad GP hyperparameters but rather from its ability to capture complex non-Euclidean input correlations. For the hyperprior, we closely follow Hvarfner et al. (2024) and define a log-normal distribution as the hyperprior for all lengthscales $\ell_i$ as

$$p(\ell_i) \sim \mathcal{LN}\left(\mu_0 + \frac{\log D}{2}, \sigma_0\right) \tag{17}$$

and set the constants to $\mu_0 = \sqrt{2}$ and $\sigma_0 = \sqrt{3}$. As in Hvarfner et al. (2024), we no longer use box constraints on the lengthscales for the $D$-scaled GP. The lengthscales are initialized with the mean of the hyperprior at each iteration and then optimized by minimizing the marginal log-likelihood. In the following, we focus only on Ackley (5D), Pestcontrol, and NNDraw. The results are in Figure 13

We can observe that for Ackley, the additional hyperprior does not have an influence on the performance. For the NNDraw example, the inductive bias introduced by the hyperprior benefits performance when using `logEI`, but it is negligible for Thompson sampling. In high-dimensional settings, the critical factor for Thompson sampling is not the kernel choice but whether the surrogate model is parametric or non-parametric. On Pestcontrol, the hyperprior not only fails to improve performance but also negatively impacts it when using `logEI`. Even with the additional hyperprior, the GP model is not able to capture the input correlations, which prevents it from yielding good

performance. Here a change in kernel is necessary to achieve competitive performance with a GP surrogate model. The VBLLs and the other parametric BNN models can capture this complex input correlation and can converge fairly quickly to the optimum, given that the problem has 25 dimensions. Lastly, it is important to note that also for the VBLLs and the other baselines, different hyperparameters such as network architecture can also improve performance (see Appendix E.3).

### C.3 COMPARISON OF SENSITIVITY TO NOISE FOR LLLA AND VBLLS

We conduct additional experiments to compare the robustness of the LLLA and VBLL surrogates to objective noise. Specifically, we consider Ackley2D and Ackley5D and set the output noise standard deviation to $\sigma \in [0.0, 0.001, 0.01, 0.1, 1]$ to test a broad spectrum. For each $\sigma$, we run 10 seeds. The results are summarized in Figure 14. The left plot displays the mean and the 10th and 90th percent quantiles across all $\sigma$. We can observe that the performance is more consistent for VBLLs on the lower dimension problem. The center and right plots show mean and quantiles for each $\sigma$ for LLLA and VBLLs, respectively. As a reference, we overlay the performance of the noise-free case in blue. Also here we can observe that while the mean of the best observed values[6] remains similar for Ackley2D, VBLLs display higher consistency compared to LLLA considering the quantiles. A similar but less amplified trend can be observed on Ackley5D.

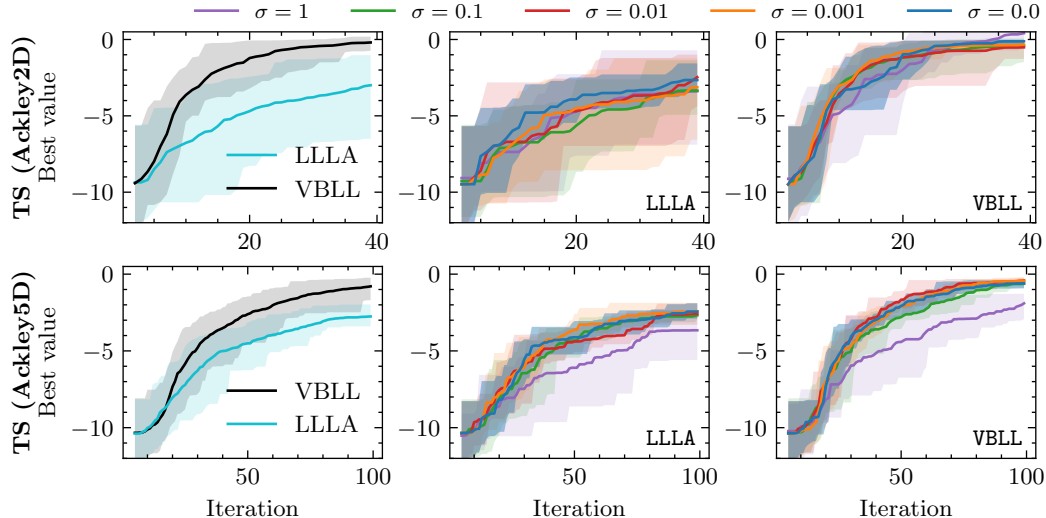

Figure 14: Sensitivity to noise on Ackley2D and Ackley5D for LLLA and VBLLs.

## D EXPERIMENTAL DETAILS

### D.1 BASELINES

**GPs:** As the de-facto standard in BO, we compare against GPs. As kernel, we choose a Matérn kernel with $\nu = 2.5$ and use individual lengthscales $\ell_i$ for all input dimensions that are optimized within box constraints following recommended best practices (Eriksson et al., 2019; Balandat et al., 2020). We use box constraints as $\ell_i \in [0.005, 4]$ (Eriksson et al., 2019)–we refer to Appendix C.2 for further discussion on this choice as well as a comparison to $D$-scaled GPs (Hvarfner et al., 2024). We expect the performance of GPs to be particularly good on stationary benchmarks.

**I-BNNs:** We compare against infinite-width Bayesian neural networks (I-BNNs) (Lee et al., 2018), which have shown promising results in recent work (Li et al., 2024). As in Li et al. (2024), we set the depth to 3 and initialize the weight variance to 10 and the bias variance to 1.6. Note that the I-BNN is expressed as a kernel and therefore the model is still non-parametric and does not learn features.

---

[6]Note that we track the best *observed* value. Therefore, values greater zero are possible (see $\sigma = 1$ for the VBLLs on Ackley2D) even though $\max_{\boldsymbol{x} \in \mathcal{X}} \mathbb{E}[f(\boldsymbol{x})] = 0$ for Ackley.

**DKL:** Deep kernel learning (DKL) (Wilson et al., 2016) combines feature extracting with neural networks with GPs. It uses a neural network $g_\theta$ as input wrapping to the kernel as $k_{DKL} \coloneqq k(g_\theta(\boldsymbol{x}), g_\theta(\boldsymbol{x}'))$ to allow for non-stationary modeling and exact inference. For the neural network, we use the same architecture as in (Li et al., 2024), i.e., 3 layers with 128 neurons. We further use ELU activations for all layers.

**LLLA:** Last layer Laplace approximations (LLLA) are a computationally efficient way obtain uncertainty estimates *after* training a neural network (MacKay, 1992; Daxberger et al., 2021). With this, they are a well suited BNN surrogate model for BO (Kristiadi et al.; Li et al., 2024). As NN, we also use 3 layers with 128 neurons and ELU activations. Note that for TS, we can also optimize the parametric neural network function numerically with LLLA.

**DE:** Deep ensembles are a cheap and efficient way to obtain predictions with uncertainty. Each member of the ensemble is initialized with a different random seed, and all are trained on the same data, minimizing an MSE loss using weight decay. We use 5 ensemble members. We parameterize the predictive as a normal distribution with mean and variance from the ensemble. We note that an ensemble of VBLL networks is also possible, although we do not include this approach.

**VBLL:** For the VBLLs, we use the same architecture as for DKL and LLLA. In the body of the paper, we include two VBLL models: training the features from scratch every iteration (performant but expensive) and re-training the features based on an event trigger and otherwise doing recursive last layer updates (VBLL (ET CL)).

### D.2 EXPERIMENTS

**Branin:** A standard two dimensional optimization benchmark with three global optima.

**Ackley:** A standard optimization benchmark with various local optima (depending on the dimensionality) and one global optimum. In out experiments, we compare the surrogates on a 2D and 5D version and set the feasible set to the hypercube $\mathcal{X} = [-5, 10]^d$ as in Eriksson et al. (2019).

**Hartmann:** A standard six dimensional benchmark with six local optima and one global optimum.

**NN draw:** In this optimization problem, our goal is to find the global optimum of a function defined by a sample from a neural network within the hypercube $\mathcal{X} = [0, 1]^d$. This benchmark was also employed in Li et al. (2024). We use a fully connected neural network with two hidden layers, each containing 50 nodes, and ReLU activation functions. The input size corresponds to the dimensionality of the optimization problem (in our case, 200), and the output size is one. To generate a function, we sample all weights from the standard normal distribution $\mathcal{N}(0, 1)$. For a fair comparison, we use the same fixed seed across all baselines ensuring that the same objective function is used.

**Pest control:** This optimization problem was also in Li et al. (2024) and aims to minimizing the spread of pests while minimizing the prevention costs of treatment and was introduced in Oh et al. (2019). In this experiment, we define the setting as a categorical optimization problem with 25 categorical variables corresponding to stages of intervention, with 5 different values at each stage. As mentioned in Oh et al. (2019), dynamics behind this problem are highly complex resulting in involved correlations between the inputs.

**Lunar lander:** Lunar lander is an environment from OpenAI gym. The objective is to maximize the average final reward over 50 randomly generated environments. For this, 12 continuous parameters of a controller have to be tuned as in Eriksson et al. (2019).

**Branin Currin:** A standard two dimensional benchmark with two objectives.

**DTLZ1:** A standard benchmark of which we choose a five dimensional version with two objectives.

**DTLZ2:** A standard benchmark of which we choose a five dimensional version with two objectives.

**Oil Sorbent:** Building on the implemention by Li et al. (2024), we optimize the properties of a material to maximize its performance as a sorbent for marine oil spills as introduced by Wang et al. (2020). The problem consists of five ordinal parameters and two continuous parameters which control

the manufacturing process of electrospun materials, and the three objectives are water contact angle, oil absorption capacity, and mechanical strength.

# E    Hyperparameter Sensitivity

The parametric VBLL surrogate has hyperparameters that have to be specified a-priori. In the following, we present results on the hyperparameter sensitivity of the VBLL surrogate model and demonstrate that tuning hyperparameters can improve empirical performance but also that the VBLL surrogate model is rather robust for a wide range of specifications. In Sec. E.1, we will first consider the sensitivity with respect to the Wishart scale and the reinitialization rate for continual learning. Following this, Sec. E.3 then studies the sensitivity regarding the width of the neural network backbone. Lastly, Sec. E.4 considers the robustness to different noise levels.

## E.1    Wishart Scale and Continual Learning Sensitivity

We sweep a number of hyperparameters in the VBLLs in order to experiment with the hyperparameter sensitivity of the VBLL models. In particular we sweep the Wishart scale and the re-initialization rate of the model. The re-initialization rate determines how often the VBLL model is re-initialized rather than using CL on the backbone and the variational posterior.

The results of the hyperparameter sweep of the Wishart scale and reinitialization rate on Ackley (5D) can be seen in Fig. 15 and on Pestcontrol in Fig. 16. Please note that we have computed running averages on these figures to make qualitative assessment easier. We find that both of these hyperparameters have impact on BO performance, but that the VBLL models are not exceedingly brittle to the values of these hyperparameters. These results also indicate that continual learning for VBLLs is an area of interest, not only to reduce fitting time, but also because there are indications that the VBLL surrogate benefits from not always being re-initialized (see e.g., Fig. 15 (b) for reinitialization rates 3 and 5). Based on these results we also hypothesise that tuning the Wishart scale appropriately for the problem at hand may lead to increased model performance.

## E.2    Influence of the Wishart Scale on the Performance of Continual Learning

We can leverage the insights from the previous section also to improve the performance of the continual learning baseline for `logEI`. We saw that decreasing the Wishart scale can improve performance. The default value of the Wishart scale used in all experiments in Section 5.2 and Section 5.3 was $V = 0.01$. This induces a hyperprior on the noise with a mass centered at $\sigma^2 = 0.01$. However, expected improvement is known to get stuck for noisy objectives. We can observe that combined with recursive updates this effect is increased compared to the standard VBLL baseline which still converges well due to the additional randomness of a new initialization (cf. Figure 3). In Figure 17, we set the Wishart scale to $V = 10^{-5}$ for the continual learning baseline. We include the standard VBLLs as a reference. We can observe that with this tuned Wishart scale, we can significantly improve the performance on benchmarks such as Pestcontrol. This further shows that similar to tuning a hyperprior for a GP, tuning hyperparameters can have a benefit.

## E.3    Model Width Ablation

To evaluate the impact of model width on the performance of both the MAP and VBLL Thompson sampling methods, we conducted a series of experiments on the `Ackley5D` and `Pestcontrol` benchmarks varying the model width.

As illustrated in Fig. 18 (a) and 19 (a), increasing the model capacity (width) of the MAP baseline results in a significant increase in variance, especially pronounced in the `Ackley5D` benchmark. This high variance suggests that the MAP method is highly sensitive to changes in model width, making it challenging to tune effectively for consistent performance across different tasks.

In comparison, the VBLL method exhibits more robustness to model capacity, as shown in Fig. 18 (b) and 19 (b). Despite increasing the model width, VBLL does not suffer from the high variance observed in the MAP baseline. Additionally, the next $x$ distance metric plots in both Fig. 18 and 19 indicates that at later BO stages the VBLL models return to exploring uncertain regions of the input

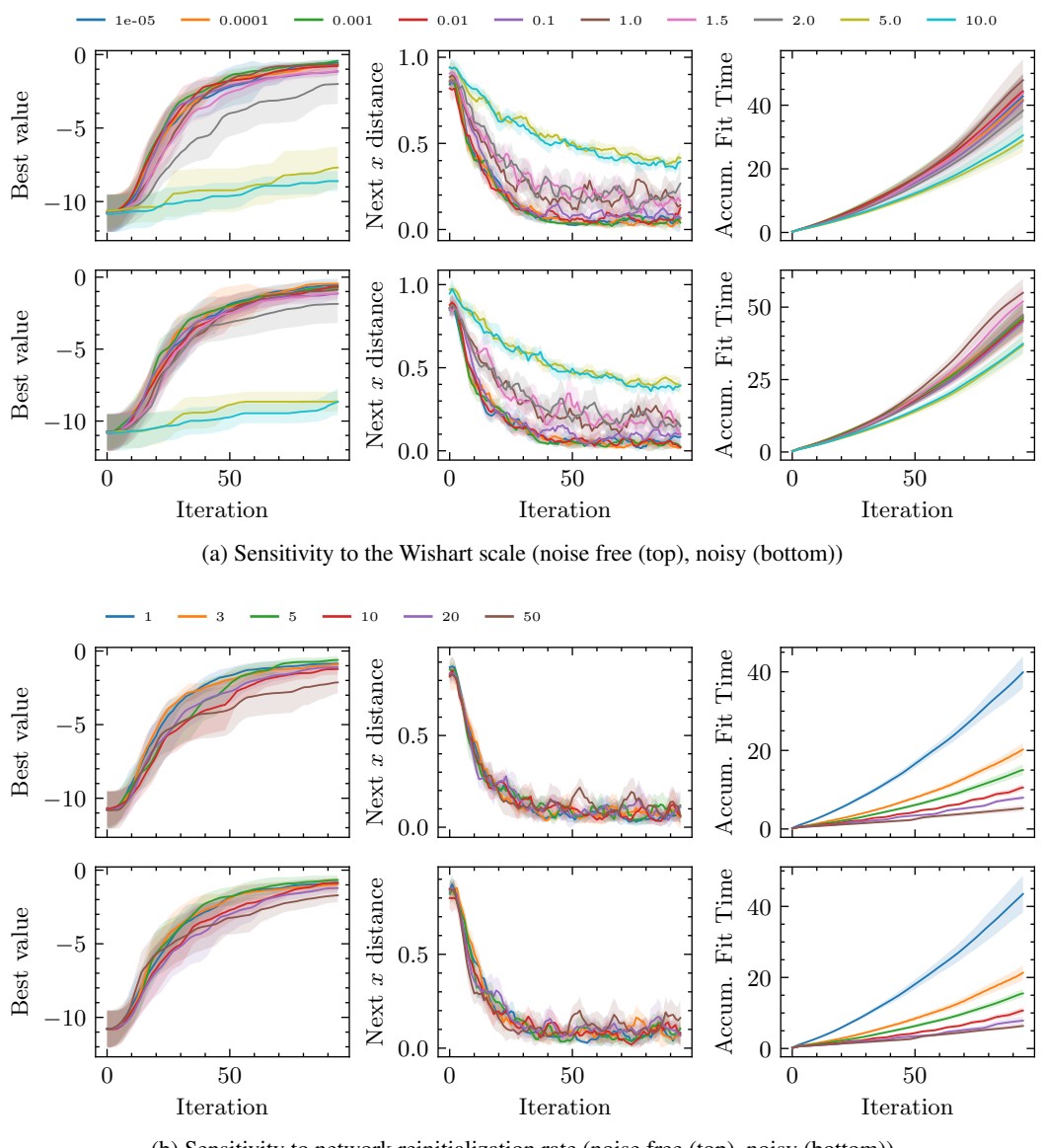

(a) Sensitivity to the Wishart scale (noise free (top), noisy (bottom))

(b) Sensitivity to network reinitialization rate (noise free (top), noisy (bottom))

Figure 15: Hyperparameter sensitivity on the Ackley5D benchmark.

space whereas the MAP models get stuck in local optima accounting for the continous improvement in best values for the VBLL models. This robustness is advantageous in practical application where extensive model tuning is unfeasible and hints that VBLL may also perform better when scaling to larger model sizes.

### E.4 MODEL PERFORMANCE IN THE PRESENCE OF NOISE

Lastly, we also benchmark the different surrogates on different noise levels. We again only consider Ackley5D (Fig. 20) and Pestcontrol (Fig. 21). For these experiments, we use the same Wishart scale of 0.01 for the VBLL baseline. We can observe that all models, besides the MAP baseline in Ackley5D, are rather robust the change in noise level.

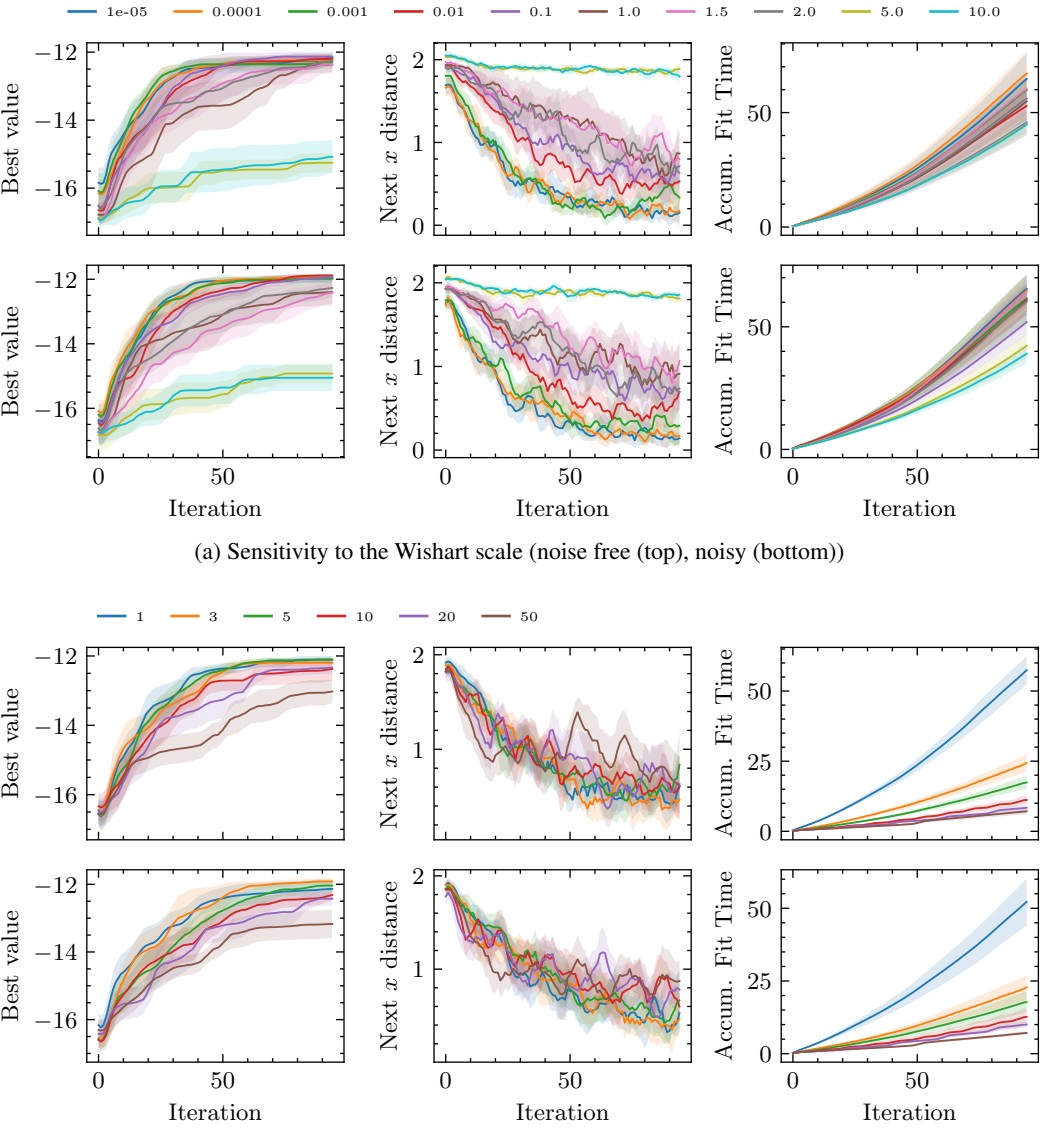

(a) Sensitivity to the Wishart scale (noise free (top), noisy (bottom))

(b) Sensitivity to network reinitialization rate (noise free (top), noisy (bottom))

Figure 16: Hyperparameter sensitivity on the `Pestcontrol` benchmark.

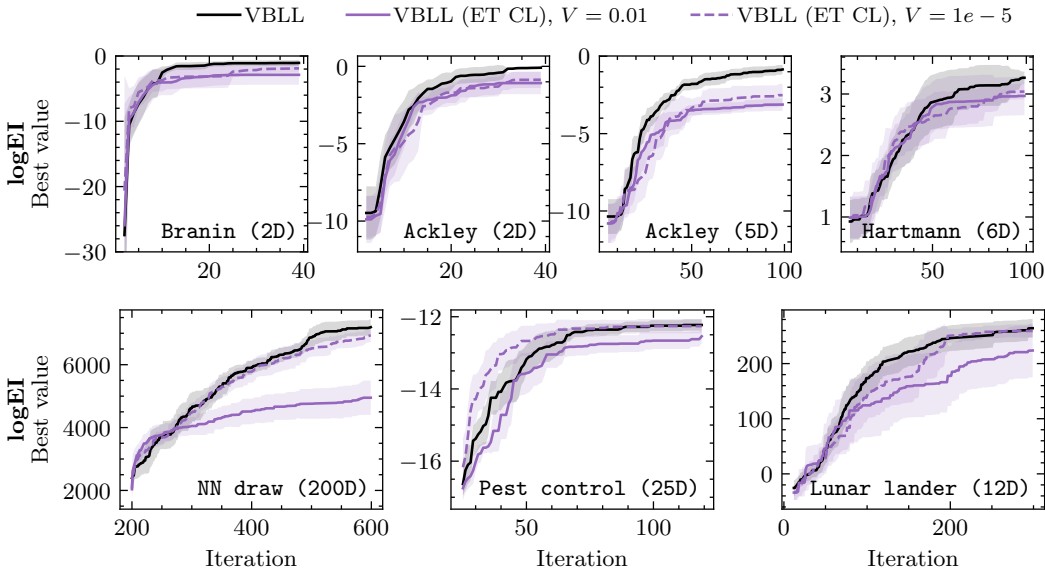

Figure 17: *Classic benchmarks (top) and high-dimensional and non-stationary benchmarks (bottom).* Performance of all surrogates for `logEI` (top) and `TS` (bottom).

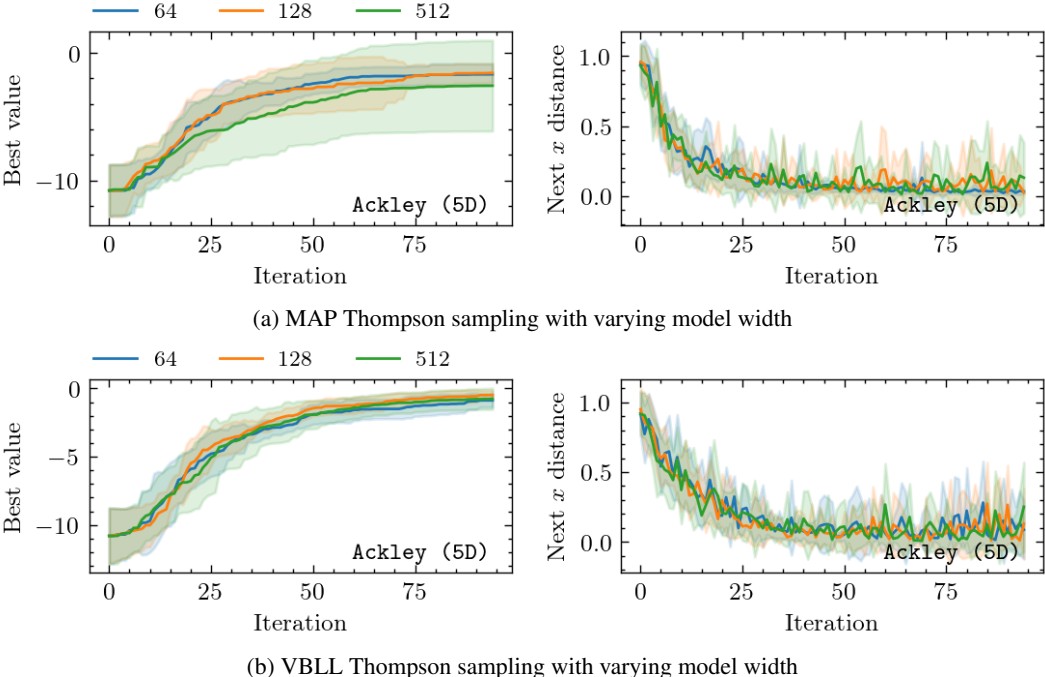

(a) MAP Thompson sampling with varying model width

(b) VBLL Thompson sampling with varying model width

Figure 18: Comparison of neural network Thompson sampling methods on the `Ackley5D` benchmark with varying model width. The models were trained with width of 64, 128 and 512 neurons.

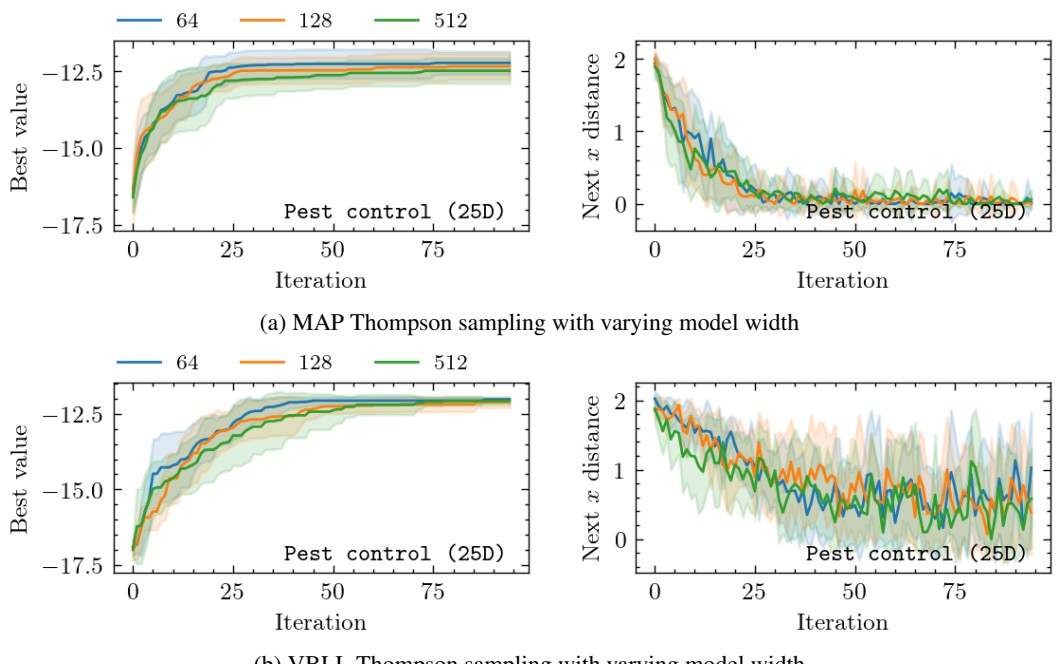

(a) MAP Thompson sampling with varying model width

(b) VBLL Thompson sampling with varying model width

Figure 19: Comparison of neural network Thompson sampling methods on the `Pestcontrol` benchmark with varying model width. The models were trained with width of 64, 128 and 512 neurons.

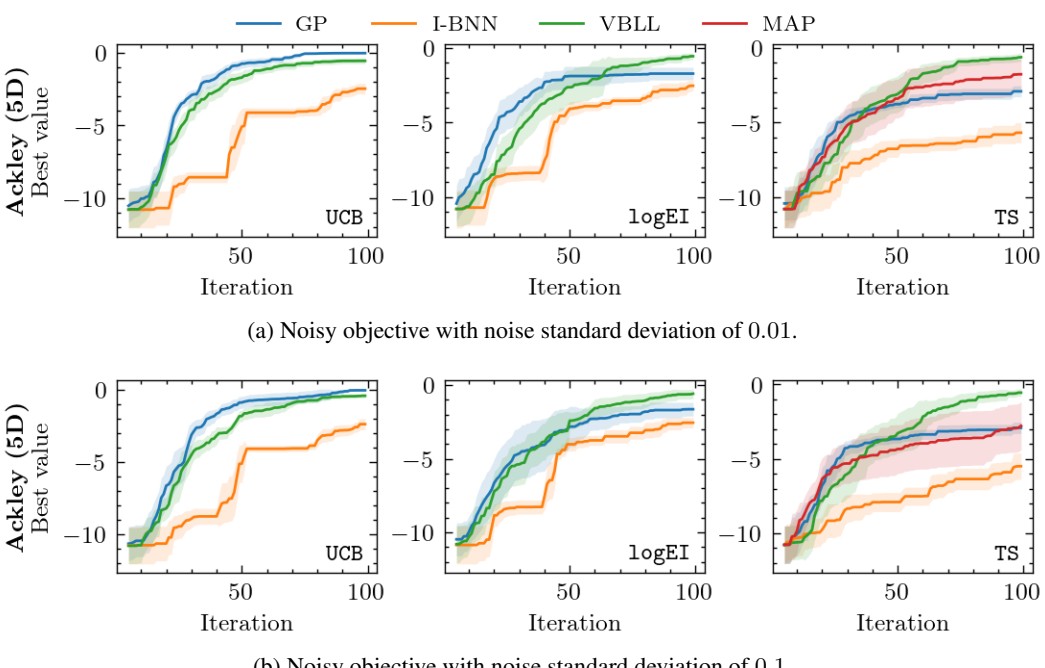

(a) Noisy objective with noise standard deviation of 0.01.

(b) Noisy objective with noise standard deviation of 0.1.

Figure 20: Performance comparison of baseline methods on `Ackley5D` benchmark with noise.

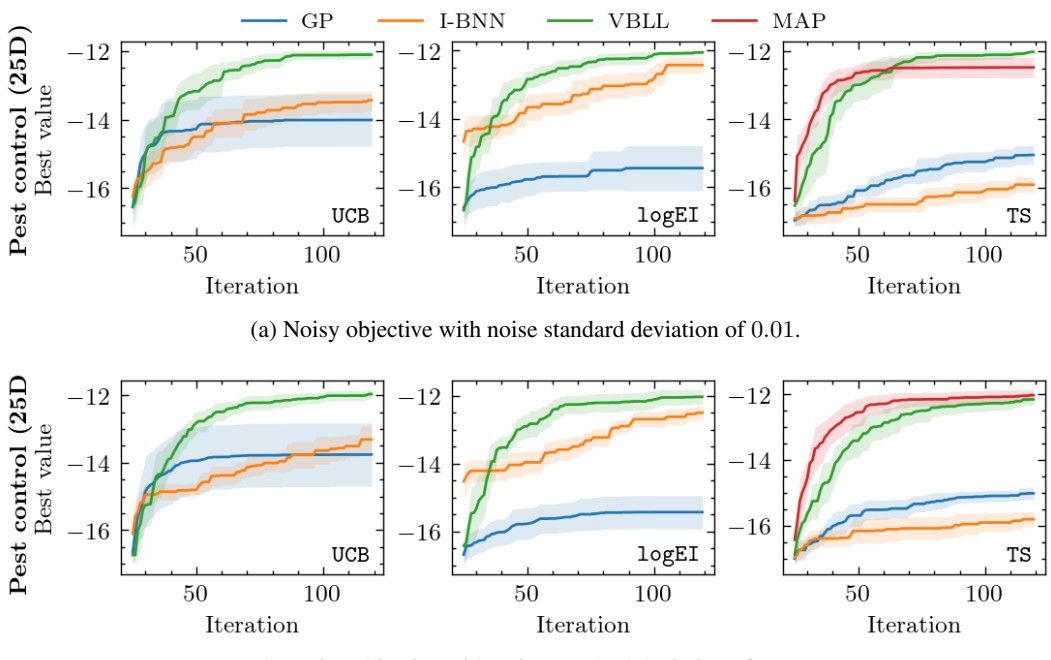

(a) Noisy objective with noise standard deviation of 0.01.

(b) Noisy objective with noise standard deviation of 0.1.

Figure 21: Performance comparison of baseline methods on `Pestcontrol` benchmark with noise.

