# OpenReview forum: "Bayesian Optimization via Continual Variational Last Layer Training"
_ICLR.cc/2025/Conference — ICLR 2025 Spotlight_

### Official Review · Reviewer_asqH · 2024-10-25

**Soundness:** 1
**Presentation:** 3
**Contribution:** 3
**Rating:** 8
**Confidence:** 4

**Summary:**

Gaussian Process (GP) models are widely regarded as state-of-the-art surrogates for Bayesian Optimization, thanks to their strong predictive performance, efficient updates for small datasets, and straightforward uncertainty quantification. However, they can be challenging to apply to search problems with non-stationary or complex correlation structures. In contrast, Bayesian Neural Networks (BNNs) are more flexible and can handle such problems with minimal adaptation, but they have traditionally been associated with high computational costs and unreliable uncertainty estimates.

This paper introduces a new method that leverages variational last-layer BNNs, combining advantages of both GPs and BNNs. The proposed approach demonstrates superior performance over GPs and several other BNN architectures in tasks with complex input correlations while achieving comparable results to GPs on standard benchmark problems.

**Strengths:**

The paper makes a valuable contribution to the field by highlighting the capabilities of a class of BNN models for Bayesian Optimization (BO) and introducing a practical technique for efficient updates in online settings, including BO scenarios. The writing is clear and well-structured, and the findings are substantiated by experiments conducted in both single-objective and multi-objective settings.

**Weaknesses:**

The presented method appears to perform comparably to Last Layer Laplace Approximations (LLLA) without clearly demonstrating new advantages. The paper emphasises computational concerns — and presumably could provide a favourable runtime-performance trade-off—but that is not empirically validated and the method is seemingly not compared against the baselines in this respect.

The claim that Laplace approximations are more sensitive to noise lacks sufficient support, as there is no accompanying experiment or reference to substantiate it (see Section 6, "Performance and Baselines"). Providing evidence here would strengthen the argument.

The discussion around early-stopping based on training loss (see Section 3.2, "Early Stopping") makes significant claims, such as training "only as long as necessary to achieve optimal performance" and suggesting that applying a similar criterion could benefit training neural network-based surrogate models in BO more broadly. While it is reasonable to argue that stopping training before full convergence improves runtime efficiency and can serve as a regularisation heuristic, the lack of experimental results of the trade-off in the setting when presented as a methodological contribution is a major omission. The effect of early stopping on the quality of the fitted model should be demonstrated through empirical evaluation, such as predictive error on relevant functions or/and assessing its impact on BO performance.

The choice of length scales [0.005, 4] for the GP model (see Section 5.1, "Surrogate Models") appears to be unsuitable for the high-dimensional benchmarks considered. As demonstrated in (1) "Vanilla Bayesian Optimization Performs Great in High Dimensions" (ICML 2024), length scales around \sqrt{D} are generally more effective for Bayesian Optimization in high-dimensional settings. Using more appropriate lengthscales (specifically adopting a suitable lengthscale prior with mass concentrated near \sqrt{D}) could potentially dramatically enhance the model's performance, making it a more informative comparison.
(1) https://arxiv.org/pdf/2402.02229

**Questions:**

1. I think the runtime advantage of the suggested algorithm must be more clearly presented, since it is the motivation of much of the methodology. Specifically, be accompanied by results showcasing a benefit (in e.g. regret/performance per wallclock time), especially compared to LLLA which in terms of regret per iteration performs similarly.

2. There are a couple of sections in the methodology which I think unjustifiably and unnecessarily make claims without backing (see Weaknesses). I recommend the authors to look over the claims and make sure they have backing; either by adding relevant proofs, experiments or references for claims important to the paper, or lessening/removing claims which may be unnecessary. It is okay that not every design decision in a larger algorithm (or system or model) is fully backed, but then those design decisions should arguably not be presented as central parts of the methodology accompanied by unbacked claims.

Overall, the paper addresses a meaningful gap in the literature. If the concerns outlined above are addressed, I would be inclined to raise my evaluation score.

---

> ### Author Response · Authors · 2024-11-25
>
> We would firstly like to thank the reviewer for the thorough comments and details. We would then like to comment on the highlighted weaknesses and address the presented questions.
>
> ---
>
> > Regarding Laplace approximations and their sensitivity to noise
>
> We agree with the reviewer that in the first version of the paper there was not sufficient evidence to support the claim on sensitivity of noise in LLLA models. However, we have now included a set of experiments in the appendix which highlight that VBLL indeed is more robust to noise than LLLA.
>
> ---
>
> > Regarding a comparison to a GP model with a D-scaled hyperprior, i.e. with mass concentrated near \sqrt{D}
>
> We now included an additional comparison to GPs with $D$-scaled hyperprior on the lengthscales in the appendix. For the construction we closely follow the suggested paper and use a LogNormal distribution with the same hyperparameters. In the results, we can observe that for the lower-dimensional problems, the influence is negligible. For the very high-dimensional NNDraw the inductive bias does help the performance when using logEI. However, for Thompson sampling there appears to be no noticeable difference. Here, the key aspect is using a parametric vs. a non-parametric model. Lastly, on Pestcontrol, a problem with complex input correlations, the D-scaled prior even not only fails to improve performance but also negatively impacts it when using logEI. This result highlights that here the type of kernel is not suitable for the problem at hand. The VBLLs (and other BNN baselines) directly learn the correlations and converge efficiently to the optimal solution and to us, these results indicate that using a D-scaled hyperprior would not alleviate the problems of GPs that are the motivation for using VBLL surrogates.
>
> ---
>
> > Regarding Question 1
>
> With regards to the comments on runtime discussion, we agree that this was lacking in the previous paper. We have included comparisons of surrogate fit times and BO performance in the revised version of the paper, and refer to the general comment for a more thorough discussion and comments on the accumulated fit times. We would also like to add that VBLL outperforms LLLA on many of the classic benchmarks (Figure 3 (top)).
>
> ---
>
> > Regarding Question 2
>
> With regards to claims without backing on early stopping, we agree with the reviewer that some phrasings and claims were too strong and will amend this in the revised version. For the early stopping, we have replaced “optimal” with “good empirical performance” the revised version. To support this claim, we have added an additional experiment in Appendix B.2 (Figure 6 (b)), showing that the performance is not significantly impacted by the early stopping, but we can significantly reduce the runtime.

---

> > ### Comment · Reviewer_asqH · 2024-11-26
> >
> > Thank you for your answers and the additional experiments you have run. I have some follow-up questions.
> >
> > **Regarding Laplace approximations and their sensitivity to noise**
> > On the two functions presented in Figure 12, VBLL and LLLA seem to me to be performing similarly in both settings of high and low noise. That is, although VBLL performed better on these functions than LLLA, both VBLL and LLLA seem largely unaffected by the noise, performing similarly with and without noise. I struggle to draw the conclusion, based on this evidence, that VBLL is  inherently more robust to noise than LLLA. Can you guide me through the interpretation of Figure 12 that supports the conclusion?
> >
> > **Regarding a comparison to a GP model with a D-scaled hyperprior, i.e. with mass concentrated near \sqrt{D}**
> > Thank you for adding this. Looking at C.2. "COMPARISON TO D-SCALED GAUSSIAN PROCESS PRIORS" I cannot find a description on how you change the initialisation and the optimisation to accommodate for the new lengthscale prior. Are you still using the constraint on the lengthscale, $l_i \in [0.005, 4]$?

---

> > > ### Author Response · Authors · 2024-11-27
> > >
> > > Thank you for engaging in the discussion! We hope to clarify the open points further.
> > >
> > > ---
> > > > Regarding Laplace approximations and their sensitivity to noise. Can you guide me through the interpretation of Figure 12 that supports the conclusion?
> > >
> > > Yes, definitely. We do generally agree with the reviewer that the effect of the noise on the final performance is not very significant in Figure 12. What we observed was that on Pestcontrol there is a trend noticeable for UCB and logEI (more noise worsens performance). We do however agree that the main message we wanted to get across was not well supported by the Figure. We now exchanged Figure 12 with additional experiments to clearly highlight the effect of noise on the LLLA and VBLL surrogate models (new Figure 12 in Appendix C.3). Specifically, we ran additional experiments on Ackley2D and Ackley5D and now tested four different noise values ($\sigma \in [1e-3, 1e-2, 1e-1, 1]$) instead of two as in the previous version of the paper. We now also show LLLA and VBLLs in separate plots (but same yaxis) to better see the trends in performance. These new results show that while the mean performance remains similar, the consistency of the VBLLs is better across runs compared to the LLLA. This is especially the case for the lower-dimensional Ackley2D. We believe that these new results give some evidence for the claim that LLLA surrogates “appear more sensitive to observation noise” (L. 533 in re-revised version) but if the reviewer does not find the evidence sufficient for the claim, we are not opposed to retracting the statement from the discussion in a final version and just include the forward reference to the experimental results on the noise sensitivity.
> > >
> > > ---
> > > > Regarding a comparison to a GP model with a D-scaled hyperprior, i.e. with mass concentrated near \sqrt{D}. Are you still using the constraint on the lengthscale, $l_i \in [0.005,4]$?
> > >
> > >
> > > Thank you for your quick response, this gave us the opportunity to add these details to the paper. We agree that this is important to include but forgot to add this detail to the revised version. We have now added this for the re-revised version. For the D-scaled prior, we no longer use the constraints on the lengthscale. Such constraints are also not mentioned in [R1] and leaving them out ensures a clear comparison to the box-constrained baseline used in, e.g., [R2] and our GP baseline. As the default in BoTorch, the lengthscales are initialized as the mean of the hyperprior and are still optimized through the MLL. Again, we have updated the accompanying text in Appendix C.2 to avoid misunderstandings.
> > >
> > >
> > > ---
> > > # References
> > >
> > > [R1] Hvarfner, Carl, Erik Orm Hellsten, and Luigi Nardi. "Vanilla Bayesian Optimization Performs Great in High Dimension." ICLR (2024).
> > >
> > > [R2] Eriksson, David, et al. "Scalable global optimization via local Bayesian optimization." NeurIPS (2019).

---

> > > > ### Author Response · Authors · 2024-12-02
> > > >
> > > > Dear reviewer,
> > > >
> > > > We are now within 24 hours of the reviewers' response deadline, and we were hoping that we could possibly address any outstanding points that the reviewer may have. We believe and hope we have addressed the weaknesses and required clarifications with the additional presented results, new baselines, and accompanying discussion. We would be grateful if the reviewer would consider changing their score given the comment in their initial review. We are happy to provide any further clarification if necessary.

---

### Official Review · Reviewer_StZZ · 2024-11-01

**Soundness:** 3
**Presentation:** 3
**Contribution:** 3
**Rating:** 8
**Confidence:** 3

**Summary:**

Although Gaussian processes (GPs) are widely used for Bayesian Optimisation (BO), they are not always well suited to modelling functions with complex correlations across inputs, and are often limited by the choice of kernel function. On the other hand, Bayesian neural networks (BNNs) can better handle complex non-Euclidean data, but are computationally expensive and challenging to condition on new data. In this work, the authors extend recent work on Variational Bayesian Last Layer (VBLL) models specifically for BO. They show how VBLLs can be adapted as efficient surrogate models for BO tasks through modified training procedures, which enable continual, online training with recursive conditioning, improving scalability. Additionally, the authors demonstrate how VBLL’s parametric structure enables effective Thompson sampling for both single- and multi-objective acquisition functions, offering more stability and numerical efficiency compared to GPs. Experiments compare VBLL’s performance against other techniques such as baseline GPs, and BNNs, and show that VBLL performs especially well on complex tasks, such as the Oil Sorbent benchmark, where other approaches struggle due to numerical instability.

**Strengths:**

The paper is well-written and a pleasure to read. The problem statement is clear from the outset, and the connections to related work are extensive. I also appreciated how the paper’s focus on practical aspects such as improving training efficiency via continual learning. Having the method implemented in BoTorch is also appealing to practitioners wanting to experiment using this method in real-world settings.

The experiments demonstrate that VBLL performs well in the targeted settings having complex input correlations and non-Euclidean structures. Showing that VBLL outperforms competing techniques on real-world datasets such as the Oil Sorbent and Pest Control datasets adds further credence to how VBLL is suited to multi-objective settings prone to numerical instability.

While the contributions may initially appear incremental, adapting VBLL to BO introduces challenges that require non-trivial solutions. The need for efficient, online training in BO necessitated the development of recursive conditioning and continual learning updates, which are distinct from standard regression tasks. Addressing the requirements of multi-objective and high-dimensional settings also required effective workarounds to address numerical stability issues.

**Weaknesses:**

1. Deep kernel learning was the first method to come to mind when reading the motivation for this work. While I appreciated its inclusion in the experimental section, I would have liked more discussion in the earlier sections on why DKL might be less ideal than VBLL. To my understanding, DKL’s computational complexity, especially in high-dimensional settings, might be a key differentiator, but additional detail on this would help clarify VBLL’s practical advantages right from the outset.
2. While the experiments are quite extensive, I would appreciate more insight on the cases where the method is expected to underperform compared to other approaches. Although dedicated experiments are provided in the supplementary material, high-level insights on possible sensitivity to hyper-parameters could also be included in the main text.

**Questions:**

Please refer to comments in "Weaknesses".

---

> ### Author Response · Authors · 2024-11-25
>
> Firstly, we would like to thank the reviewer for the thorough review. We would then like to address the presented weaknesses brought by the reviewer.
>
> ---
>
> > Regarding Question/Weakness 1
>
> With regards to why VBLL might be more suited than DKL: the primary reason that DKL is challenging is due to the computational complexity of gradient computation with a marginal likelihood objective, as stated by the reviewer. This becomes especially problematic for larger datasets, which was one of the motivations for the VBLL method. We have highlighted this in the appendix of the paper and tried to clarify elements and differences of the various methods presented in this work. Furthermore, DKL still yields a non-parametric model which can become problematic for high-dimensional Thompson sampling (see eg. NNDraw with TS). Essentially one can interpret the VBLLs as a parametric version of DKL.
>
> ---
>
> > Regarding Question/Weakness 2
>
> With regards to insights on where the method is expected to underperform we mainly see two areas where we suspect other surrogates may be preferred to VBLLs. The first is in settings where data is extremely scarce i.e. where total number of samples is <<100 - here it is likely that methods such as GPs are still dominant. Secondly, we still see that in comparison to some other surrogate types, the computational costs of VBLLs are quite high, and therefore in settings where surrogate model fit times are a constraint (which is not very common in BO, but does occur), other surrogate models may be more appropriate.
>
> Based on the reviewer’s recommendation, we have added some high-level comments on sensitivity to hyperparameters to the main body text but kept the results in the appendix due to a lack of space.
>
> Finally we would like to thank the reviewer for their thorough comments and hope that they have found we satisfactorily have answered their questions.

---

> > ### Comment · Reviewer_StZZ · 2024-11-26
> > **Acknowledgement**
> >
> > Thank you for your responses and for preparing the updated version of the paper. I still consider this paper to be in a good state for publication, and will be leaving my score unchanged for the time being.

---

### Official Review · Reviewer_1qWo · 2024-11-03

**Soundness:** 4
**Presentation:** 3
**Contribution:** 3
**Rating:** 8
**Confidence:** 4

**Summary:**

This paper presents a combination of two ideas: (!) Bayesian last-layer training of neural networks with (2) using a parametric Bayesian linear model for black-box function optimization. Using natural parameterization of the last layer Gaussian and assuming independent Gaussian noise, it is possible to use a continual update which is only $O(N^2)$ in the last-layer number of neurons $N$. As with every parametric Bayesian function model, the acquisition function can be directly and analytically optimized using Thompson sampling. The empirical results on a wide variety of Bayesian Optimization tasks are promising.

**Strengths:**

The paper combines two well studied ideas in an elegant way and the presentation is (relatively) easy to follow (though I would have wished a bit more emphasis on the natural parameterization of the Normal distributions as this is key to the computational efficiency). The empirical studies are extensive and well discussed.

**Weaknesses:**

One aspect that is disregarded by the paper is how to chose the network architecture for all but the last-layer; I have no idea how sensitive the quality of the proposed approach is to this. In essence, the complexity of choosing a kernel function for GPs has been shifted to the network architecture of the underlying neural network. This is not discussed in sufficient detail. Also, only at the end the difference to Laplace approximation of the last layer is discussed; I would have expected this in the Related Work section.

**Questions:**

* In line 110 und 111 it helps to point out that $\mathbf{w} = S^{-1} \mathbf{q}$ is the vector of precision-means (for those unfamiliar with natural parameters of the Normal distribution)
* The proof of Theorem 1 in the appendix relies on the simple observation that the approximating family contains the true distribution. I would have preferred to see this in the main body of the text; it's less "mechanical" than I expected and key to the reasoning of the paper (line 178-189 can be significantly shortened b/c it uses the well known Cholesky decomposition for approximating the inverse and log-determinant of the precision)
* Line 217: Where is the parameter $V$ (Wishart prior scale) necessary?

---

> ### Author Response · Authors · 2024-11-25
>
> Thank you for your review. In the following, we hope to address all your questions and the weaknesses you highlight.
>
> Related to the comment on how neural network architecture affects the performance of the presented methods, we agree that this is a highly interesting aspect of the presented methods - we did in fact run experiments to check how neural network width affected the VBLL performance, and found them to be highly robust to varying widths (these results are currently in the Appendix for space reasons). More complex changes in neural networks architecture such as changing of architecture type is a highly interesting research direction, but requires careful selection of datasets, architectures and tuning and we therefore chose not to perform such experiments here. We hope the reviewer agrees that showing that the VBLL methods are robust to neural network width changes provides evidence that these methods are not brittle to architecture changes, which we of course agree is immensely important.
>
> Additionally, we would like to add that we agree that the placement of the comparison of VBLL and Laplace was in an odd place, but have chosen to move it to the appendix due to space constraints, but refer to it already in the Related Work and Background section.
>
> ---
>
> > Regarding Question 1
>
> Thank you for the suggestion to point out that $w=S^{−1}q$ is the vector of precision-means! We have now added this to the revised version.
>
> ---
>
> > Regarding Question 2
>
> With regards to the comments on the proof of Theorem 1: We are highly constrained on space due to new results being added to the body of the paper. While the proof is indeed quite simple, the technical details take up enough space that including the full proof is challenging. We will include a brief proof sketch (that basically states exactly what the reviewer said) in the body of the paper.
> On the complexity results: we agree that the bulk of the complexity results are standard. However, we want to highlight the complexity due to the (as far as we know, novel) use of the efficient rank-1 Cholesky updating approach. We will include a brief description of this feature in the body and move most of the complexity discussion to the appendix to save space.
>
> ---
>
> > Regarding Question 3
>
> With regards to the question on the necessity of the Wishart prior scale: Thank you for pointing this out! The Wishart prior scale is used in the initialization of the model. Here, we don’t pass the noise variance, as stated in the algorithm, but the Wishart prior scale. We learn the noise online alongside the variational posterior over W. We have fixed this typo in the revised version.
>
> Finally we would like to thank the reviewer for their time and thorough comments.

---

### Official Review · Reviewer_wtu3 · 2024-11-03

**Soundness:** 2
**Presentation:** 3
**Contribution:** 2
**Rating:** 5
**Confidence:** 4

**Summary:**

The authors propose VBLL networks as a surrogate for Bayesian optimization and identify a relationship between optimizing the variational posterior and recursive Bayesian linear regression. They demonstrate competitive BO results with VBLL on diverse single and multi-objective benchmarks.

**Strengths:**

- The method is well-explained and is theoretically justified, and there are additional modifications which can be made to increase the efficiency such as feature re-use and sparse full model retraining. This flexibility enables practitioners to balance the tradeoff between model performance and computational cost.
- The authors use a diverse setting of test objectives, specifically demonstrating performance on instances with high-dimensionality and non-stationarity.
- VBLL appears to be robust to hyperparameter choices and can be used as a drop-in surrogate model, unlike typical GPs which require careful kernel selection.

**Weaknesses:**

- Although it appears that one of the primary motivations behind this work is the increased efficiency compared to other BNN surrogates, there is no measure of runtime or computational cost within the paper. It would be helpful to understand how these methods perform as a function of computational budget. This could also help clarify the difference in performances between VBLL and VBLL CL.
- There is also currently no demonstration of why this approximation would be preferred over the using the exact marginal likelihood with approaches like DNGO [1]. Without these baselines, there is minimal evidence that the proposed method has practical merit over existing work. Furthermore, it would also be useful to compare last-layer methods like VBLL to more expensive BNN surrogates like deep ensembles so we can assess the tradeoff between computation and performance.

[1] Snoek et al, Scalable Bayesian Optimization Using Deep Neural Networks, 2015

**Questions:**

- Could you further elaborate on the differences between LLLA and VBLL? LLLA outperforms VBLL on many of the non-stationary benchmarks, and from line 530, it appears that VBLL-based approaches may be less flexible than last-layer Laplace approximations.

---

> ### Author Response · Authors · 2024-11-25
>
> Thank you for the thorough review. In the following, we aim to address all the mentioned weaknesses.
>
> ---
>
> > Regarding Weakness 1
>
> We thank the reviewer for the comments related to missing measurements of runtime and computational cost and agree that this indeed would improve the paper. We have now included figures where we plot accumulated surrogate fit times versus BO performance - see general comment for more discussion on this. As outlined in the general comment, we would like to underline that we did mean to claim that VBLL models are always computationally more efficient than other BNN surrogates. Rather, we show that VBLL models often outperform other BNN surrogates especially in the synthetic setting (especially last layer versions), yet can be costly to fit in the BO scheme. To alleviate this, we propose the recursive update scheme presented in the paper which is novel for these model types as well as continual learning scheduling that spares computational time with little cost to BO performance.
>
> ---
>
> > Regarding Weakness 2 and Question 1
>
> With regards to comparing to more expensive BNN surrogates, we have now included Deep Ensemble (DE) experiments on the single-objective problems. We generally observe that DEs perform well on the structured problems (on par with VBLL), but VBLLs generally outperform DEs on many of the synthetic problems, e.g., Branin.
>
> With regards to comparison to DNGO, the methods presented in [1] are quite similar to LLLA. In [1] the authors train a neural network end-to-end to learn the basis functions (including with a final linear layer) using a MAP estimate, and after convergence replace the linear layer with a Bayesian linear regressor. In LLLA, the full network is similarly trained end-to-end in traditional MAP fashion and after convergence, a Gaussian approximation is placed on the final linear layer parameters (keeping the MAP estimate as the mean of that Gaussian). In essence, the main difference between these two methods is that the mean of the MAP estimate is replaced with the Bayesian linear regression mean in the DNGO method, whereas the MAP estimated mean is kept in LLLA. The computation of the variances in each method are the same [3, Appendix B.1.2]. In addition, others have observed that when DNGO or BLL models are fitted in this fashion of MAP estimation followed by a replaced final layer, the learned features are not fit to provide good uncertainty estimates which cannot be amended by simply replacing the final linear layer with a Bayesian linear regressor (see [2] for details) - VBLLs should in principle not struggle with this problem since features and variational parameters are jointly learned. It is perhaps also for this reason that BLL models have not seen much popularity in recent years: see eg [4], which also does not compare to BLLs in Bayesian Optimization.
>
> The reviewer also commented that VBLL may be less flexible than LLLA, but we would argue the contrary for a number of reasons such as being able to optimize the noise covariance and jointly learning features with the variational distribution in VBLL, which may alleviate problems such as those described in [2].
>
> We hope the reviewer has found that we have satisfactorily addressed their main concerns of the paper. Should this not be the case we are happy to further address any other concerns.
>
> ---
>
> ## References
>
> [1]: Snoek, Jasper, et al. "Scalable bayesian optimization using deep neural networks." International conference on machine learning. PMLR, 2015.
>
> [2]: Ober, Sebastian W., and Carl Edward Rasmussen. "Benchmarking the neural linear model for regression." (2019).
>
> [3]: Daxberger, Erik, et al. "Laplace redux-effortless Bayesian deep learning." Advances in Neural Information Processing Systems 34 (2021): 20089-20103.
>
> [4]: Li, Yucen, et al. “A Study of Bayesian Neural Network Surrogates for Bayesian Optimization.” International Conference on Learning Representations (2024).

---

> > ### Comment · Reviewer_wtu3 · 2024-12-02
> >
> > Thank you for your response! I have updated my score accordingly.

---

### Author Response · Authors · 2024-11-25

# General rebuttal

We would like to thank all of the reviewers for their thorough comments and reviews. In this general comment, we describe an updated continual learning scheme that significantly improves not only VBLL surrogate fit times but also BO performance at a much lower cost than the one in the first version of the paper. We also present detailed comments and discussions on points brought up by two or more reviewers.

## Paper amendment: Updated continual learning schemes
In our initial submission, the continual learning methods consisted of periodically alternating between re-initializing the VBLL model and performing continual learning through recursive updates at a fixed rate to save computational resources. In the new version of the paper we have amended this periodic-update approach to an event-triggered approach which significantly improves the continual learning performance whilst reducing surrogate fit times for the VBLL models. We outline the approach here but it can also be found in the revised paper.

The event-triggered approach considers the log-likelihood of the incoming new data given the current model and compares it to a threshold. Intuitively, if the likelihood of the incoming data is high, a recursive update is sufficient as it is in accordance with the current posterior predictive whereas if the likelihood is low, the model should be re-initialized to yield better basis functions for the task at hand.

We find that the event-triggered approach performs very strongly especially in combination with Thompson sampling, nearly matching a retrain-every-iteration VBLL model at a smaller cost. We include and compare performance of the continual learning methods, i.e. periodic reinitialization as in the first version of the paper and event-triggered reinitialization, in the revised version in Appendix B.3 (Figure 7).

(continue in next comment)

---

> ### Author Response · Authors · 2024-11-25
>
> (continuation form previous comment)
>
> ## Regarding wall-clock comparisons of methods:
>
> A couple of reviewers brought up that the paper in its current version does not display any wall-clock or computational costs. We agree that this was lacking, and have now included a figure in which we plot accumulated surrogate fit time versus BO performance, firstly showcasing the fact that VBLLs are expensive to fit, but have high data efficiency, and secondly that the proposed CL scheme improves fit times.
>
> One thing we would like to note is that the focus of this paper is not to claim that VBLL models are extremely efficient in terms of surrogate fit times. One of the four primary contributions is that we showcase that VBLLs as surrogates perform similarly to GPs in the synthetic setting (which is not a small feat, see e.g. [1, Figure 3 top] where no BNN surrogate consistently performs similarly to GPs on standard benchmarks) and moreover that they outperform GPs in non-euclidian problem settings, whilst performing similarly or better than other BNN surrogates. However, the VBLLs were often expensive to fit. Another of the primary contributions is therefore the proposed recursive update and continual learning scheme to alleviate these fit times with little cost in terms of BO performance. We acknowledge that the abstract previously could have been interpreted as VBLLs being cheap to fit, and we have therefore made amendments to it.
>
> Finally, for the reviewers’ curiosity, we would like to note that we suspect that the VBLL surrogate fit times currently are dominated by the variational posterior parameter optimizations. In the problem settings in the paper, the neural networks are (relatively) small networks, and therefore the overall network fitting is dominated by the variational posterior optimizations. In settings where the non-variational parameters of the network grow (such as with large neural networks), we suspect that the gap between VBLL fit times and other BNN methods should drop. To be more precise, consider a neural network with $n$ weights in the last layer, and $m$ weights in the remaining network. Due to the parametrization of the variational posterior we have $n + 0.5n^2$ parameters for the last layer and $m$ parameters for the feature mappings. In larger networks where $m$>>$n^2$, we thus suspect the gap in surrogate fit times for the BNN surrogates should drop. In our BO setup for Ackley (5D) for example,  we have $m\approx33.800$ and $n^2\approx16.600$, i.e. the variational parameters make up $33\\%$ of the total number of parameters. In contrast, PaiNN networks [2], a type of graph network often used for regression tasks on chemical compound spaces, would have $m \approx 535.000$ and $n^2\approx 65.500$, meaning the variational parameters would make up approximately $10\\%$, whilst in MACE [3] models for materials discovery, the variational parameters would make up approximately $1.5-6\\%$ of the total parameters depending on how the final layer size was chosen.
>
> ---
> # References
>
> [1] Li, Yucen, et al. “A Study of Bayesian Neural Network Surrogates for Bayesian Optimization.” ICLR, 2024.
>
> [2] Schütt, Kristof, et al. "Equivariant message passing for the prediction of tensorial properties and molecular spectra." ICML, 2021.
>
> [3] Batatia, Ilyes, et al. “A foundation model for atomistic materials chemistry.” ArXiv, 2023.

---

### Meta-Review · Area_Chair_MXwc · 2024-12-19

**Metareview:**

The paper 'Bayesian Optimization via Continual Variational Last Layer Training' was reviewed by 4 reviewers who gave it an average score of 7.25 (final scores: 5+8+8+8). The paper has multiple strengths in presentation, how the approach is set up, and experiments. Several reviewers pointed out that the method itself appears straightforward, but there is actually more than meets the eye in getting it working. The reviewer consensus is to accept this work.

**Additional Comments On Reviewer Discussion:**

The authors posted rebuttals and the majority of the reviewers interacted during the author-reviewer discussion phase. After the internal (reviewer-AC) discussion, all reviewers support accepting this work (even if not all scores are updated).

---

### Decision · Program_Chairs · 2025-01-22

Accept (Spotlight)